# Discovering type I cis-AT polyketides through computational mass spectrometry and genome mining with Seq2PKS

Donghui Yan[1,7], Muqing Zhou [1,7], Abhinav Adduri[1,7], Yihao Zhuang[2], Mustafa Guler [1], Sitong Liu[1], Hyonyoung Shin [1], Torin Kovach[1], Gloria Oh[1], Xiao Liu [1], Yuting Deng[1], Xiaofeng Wang[3], Liu Cao[1], David H. Sherman [3,4], Pamela J. Schultz[2,4], Roland D. Kersten [3], Jason A. Clement[5], Ashootosh Tripathi [2,3,4] ✉, Bahar Behsaz[1,6] ✉ & Hosein Mohimani [1] ✉

Type 1 polyketides are a major class of natural products used as antiviral, antibiotic, antifungal, antiparasitic, immunosuppressive, and antitumor drugs. Analysis of public microbial genomes leads to the discovery of over sixty thousand type 1 polyketide gene clusters. However, the molecular products of only about a hundred of these clusters are characterized, leaving most metabolites unknown. Characterizing polyketides relies on bioactivity-guided purification, which is expensive and time-consuming. To address this, we present Seq2PKS, a machine learning algorithm that predicts chemical structures derived from Type 1 polyketide synthases. Seq2PKS predicts numerous putative structures for each gene cluster to enhance accuracy. The correct structure is identified using a variable mass spectral database search. Benchmarks show that Seq2PKS outperforms existing methods. Applying Seq2PKS to Actinobacteria datasets, we discover biosynthetic gene clusters for monazomycin, oasomycin A, and 2-aminobenzamide-actiphenol.

Natural products are important sources for drugs, and currently, ~51% of clinically approved small molecules are inspired or derived from natural products, or unaltered natural products[1]. Polyketides are a significant class of metabolites that are used as antiviral[2], antibiotic[3,4], antifungal[5], antiparasitic[6], immunosuppressive[7], and antitumor[8] drugs. For example, ivermectin, a derivative of avermectin, was recently approved for treating river blindness. Ivermectin's discovery led to the 2015 Nobel Prize in Physiology or Medicine for its discoverers. The growing resistance to certain anti-infective drugs is a growing threat to human health, highlighting the urgent need to discover and develop compounds. Discovering polyketides can assist in the treatment of future epidemics[9], parasite outbreaks[10], and the emerging anti-microbial resistance crisis[11].

Modular Type I Cis-AT polyketides (MT1PK) are a major subtype of polyketides, mainly produced by bacteria. They include antibiotics such as rifamycin[12], antifungals such as amphotericin B[13], and anticancer agents such as rapamycin[14]. In microbial genomes, polyketide biosynthetic machinery for a given polyketide is a specific set of enzymes whose coding genes are co-located in a region called a biosynthetic gene cluster (BGC). A modular Type I Cis-AT polyketide synthase (MT1PKS) consists of various modules, each responsible for synthesizing a specific chemical substructure in the core structure of an MT1PK. Each module comprises an acyltransferase (AT) domain responsible for recruiting initial substrates to an acyl carrier domain (ACP), and other domains within each module can further modify this substrate to form the mature substructure. Substructures attached to

[1]Computational Biology Department, School of Computer Science, Carnegie Mellon University, Pittsburgh, PA, USA. [2]Natural Products Discovery Core, University of Michigan, Ann Arbor, MI, USA. [3]Department of Medicinal Chemistry, University of Michigan, Ann Arbor, MI, USA. [4]Life Sciences Institute, University of Michigan, Ann Arbor, MI, USA. [5]Baruch S. Blumberg Institute, Doylestown, PA, USA. [6]Chemia Biosciences Inc, Pittsburgh, PA, USA. [7]These authors contributed equally: Donghui Yan, Muqing Zhou, Abhinav Adduri. ✉e-mail: ashtri@umich.edu; bahar@chemia.ai; hoseinm@andrew.cmu.edu

ACPs of neighboring modules are connected by ketosynthase (KS) domains in a modular assembly pathway, where starting from the substructure produced by the first module, each subsequent module extends the polyketide chain by adding a substructure and passing it to the next module as ACP-tethered intermediates. The last module contains a thioesterase (TE) domain that releases the polyketide core structure from the MT1PKS as a cyclic or linear product. Tailoring enzymes then catalyze post-assembly modifications of the MT1PKS product into a mature bioactive molecule[15,16].

Based on this biosynthetic logic, predicting the chemical structure of polyketides from their BGCs is a challenging task that involves (a) annotating PKS domains and enzymes, (b) predicting substrates from AT domains and other domains present in each module, (c) predicting the order of substrates in the polyketide assembly pathway, and (d) applying corresponding post-assembly modifications to the polyketide core structures based on identified modification genes. Over the past two decades, various methods have been developed for polyketide genome mining and chemical structure prediction. However, existing methods are not able to accurately predict the structure of mature polyketides from BGCs due to the complex post-modifications. Notably, methods like PRISM[17] and the Genomes to Natural Products (GNP) platform[18] generate a large set of potential structures from PK BGCs, and then map them to mass spectrometry data. However, these methods require full structural information of the characterized polyketide metabolites.

The existing approaches for predicting the substrate specificity of AT domains in Cis-AT PKSs are based on analyzing an amino acid motif adjacent to each domain's active site (referred to as signatures)[19]. These methods mainly utilize nearest neighbor search or support vector machines to predict the specificity of domains by training on domains with known specificity. While these approaches outperform traditional methods based on the global homology of catalytic domains, they fail to classify ones that lack close homology.

During the biosynthesis of Cis-AT polyketides, the arrangement of genes in the assembly pathway often deviates from their linear order (e.g., non-colinear) on the polyketide synthase (PKS) BGC. The state-of-the-art SBSPKS method introduced by Khater et al.[20] utilizes reference docking domain structures and calculates the raw affinity score for pathway predictions. This approach suffers from several limitations: (i) the algorithm assumes that the structure of the reference docking domains can be applied to other docking domains, which is often not the case, as demonstrated by several studies[21,22]; (ii) this method results in a large number of tied ranks in the prediction results, complicating the selection of top candidates; (iii) the output of the method is a raw score, making its reliability difficult to interpret. Also, these methods overlook the fact that the gene order in BGCs is typically mostly preserved in the final polyketide assembly pathway. For instance, among the 27 non-colinear BGCs of type I polyketide synthases (T1PKS) examined in MiBiG, 19 differ from the genomic order by merely one gene.

Finally, predicting the function of tailoring enzymes and applying the post-assembly modifications governed by these enzymes is a crucial step for accurately predicting the structure of mature polyketides from BGCs. However, current tools such as PRISM[17] only account for modifications responsible for the cyclization of the constructed core structures, omitting other complex modifications for the sake of simplicity. Consequently, this method fails to accurately predict compounds that closely resemble the correct compounds among T1PKS-BGCs in MIBIG. Prior methods such as SBSPKS[20] and SEARCHPKS[23] construct a database of modification domains and their corresponding tailoring modifications for predicting the core structure of polyketides. In that case, the drawback is that the database size is limited.

Here, we introduce Seq2PKS, a machine learning method to streamline the process of polyketide discovery based on large mass spectral and genomics datasets collected from various microbial isolates. Seq2PKS improves the domain specificity prediction by using an extra-tree-based classification algorithm, which can classify domains that are not a close homolog of any known domain. In addition, Seq2PKS distinguishes itself by utilizing a rule-based methodology that effectively considers the complex transformations performed by other domains within each module to generate precise predictions of chain elongation structures. Moreover, Seq2PKS uses an approach for predicting the assembly pathway of Cis-AT polyketides by incorporating the gene order in the genome. Compared to other methods, the Seq2PKS method used for assembly pathway prediction includes 90 training samples in the database, which allows the algorithm to accommodate different docking domain structures. Furthermore, the algorithm assigns a probability to each order, which avoids ties and provides interpretability.

Seq2PKS enhances structure prediction reliability by incorporating a more comprehensive tailoring modification database and uses a graph-based approach for searching reaction sites and applying modifications to the constructed core structure of mature compounds. In contrast to the existing methods that predict a single molecular structure per BGC, resulting in low accuracy[17,24], Seq2PKS generates up to millions of molecular structures per BGC and then ranks those structures by searching against paired mass spectrometry data, allowing for enhanced accuracy and comprehensive analysis of the polyketide structures. In addition, Seq2PKS uses a variable mass spectral database search to identify polyketides even when a mispredicted chain extension substrate or modification exists.

## Results

### Seq2PKS overview
Figure 1 illustrates the Seq2PKS pipeline, which includes the following steps described in the method section: (a) annotating polyketide domains and enzymes by genome mining, (b) predicting substrate specificity of domains, (c) predicting the assembly order of catalytic steps for building the core structure, (d) incorporating post-assembly modifications, and (e) searching mature structures against mass spectral data using Dereplicator+[25].

### Datasets
Datasets of 624 Cis-AT domains with known substrate labels from polyketide BGCs[24] were used to train the substrate specificity prediction models. 175,201 unlabeled AT domains from antiSMASH-DB[26] were used to further refine the model. A dataset of 90 Cis-AT BGCs with annotated pathways from Blank et al.[27] was employed to train the assembly order prediction model. To compare the performance of Seq2PKS with other tools, the chemical structure of 80 Cis-AT polyketides along with reported Tanimoto similarity was obtained from the literature search.

### Benchmarking the accuracy of substrate specificity prediction
For each AT domain in our training data, we extracted 24 active site residues reported by Yadav et al.[19] by aligning the protein to a reference using MUSCLE[28] sequence alignment. These residues were then one-hot encoded as the input for various machine learning models.

To compare the prediction ability of each model when the testing samples are different in the degree of dissimilarity with the training data, we split the test data points into bins. The notation $Bk+$ represents the bin containing data points that are at least $k$ Hamming distance away from any training data points. To compare different machine learning algorithms for Cis-AT domain specificity, we conducted fivefold cross-validation using the 624 labeled training data and reported the average resulting accuracy (Fig. 2). While most algorithms exhibit similar accuracy in the overall category, the extra-tree algorithm demonstrates better generalization (i.e., it attains higher accuracies for bins that are far away from training domains based on Hamming distance). The evaluation metrics are shown in Supplementary Fig. 1. Generalizability is an essential requirement for

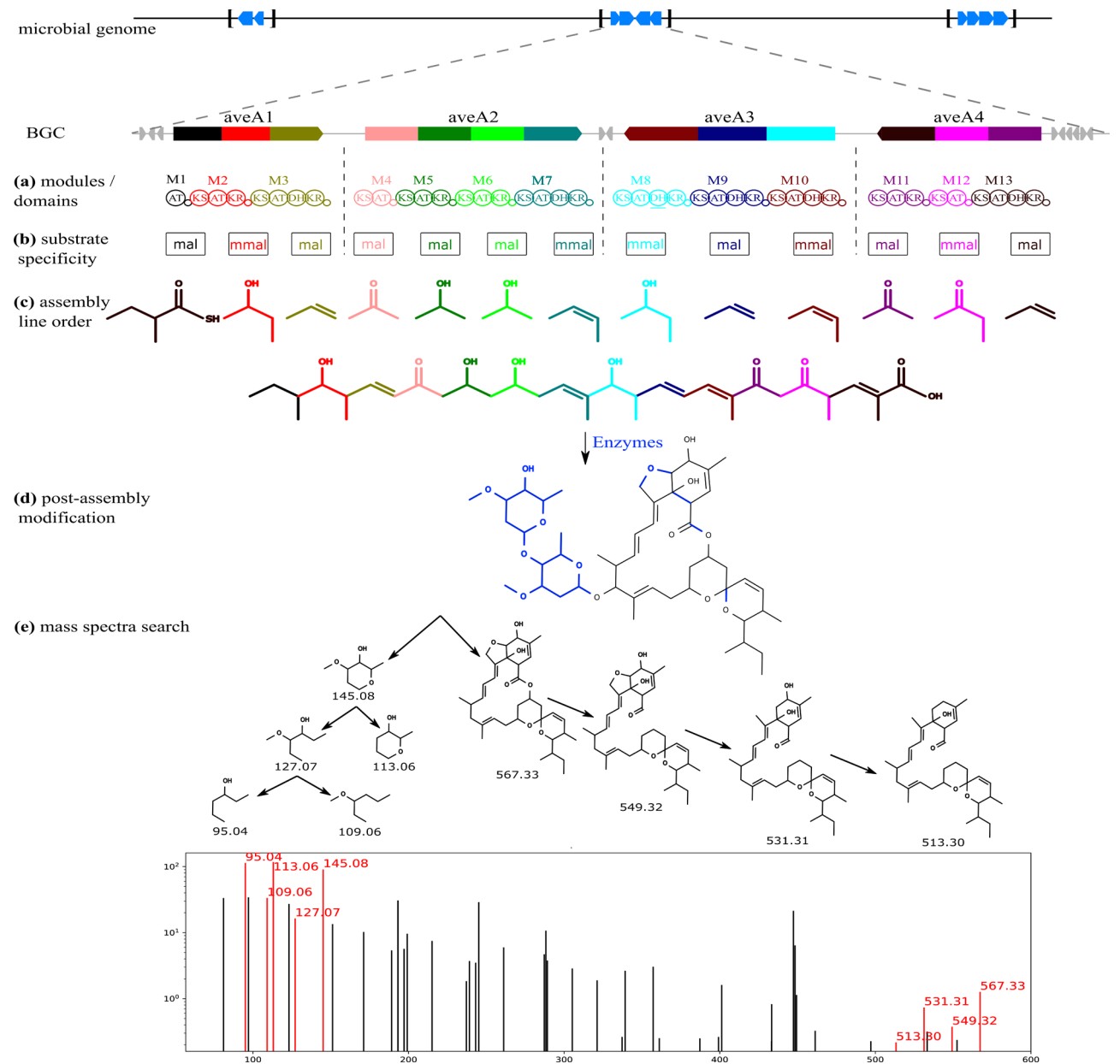

**Fig. 1 | The Seq2PKS framework.** This process initiates with the microbial genome, where (**a**) polyketide domains and enzymes are identified through genome mining (M1-M13 Modules, AT acyltransferase, KS ketosynthase, DH dehydratase, KR ketoreductase). Subsequently, **b** the specificity of the Cis-AT domains is predicted, and the corresponding substrates are determined by combining these specificities with other domains in each module (mal: malonyl-CoA; mmal: methylmalonyl-CoA). Following this, **c** the order of the assembly pathway is predicted to form the initial core structure. This leads to (**d**) the incorporation of post-assembly modifications. Lastly, **e** the mature structures are compared against mass spectra using Dereplicator+. The arrows illustrate the fragmentation process of the target molecule, and peaks in the mass spectra corresponding to these fragments are highlighted in red[25].

analyzing BGCs, as the signatures of unlabeled AT domains are widely variable, and a large portion of them are significantly distinct from the training (labeled) data (Supplementary Fig. 2).

Seq2PKS utilizes a rule-based technique to predict the structure of mature PK substrates (Supplementary Fig. 3). This prediction relies on the specificity of the AT domain and the functionality of other domains within each biosynthetic module, which has been extensively discussed in prior work[20,24]. Seq2PKS also identifies inactive domains and takes them into account for accurate prediction of the mature substrates.

## Benchmarking the accuracy of assembly pathway order prediction

In contrast to colinear Cis-AT PKSs, many non-colinear systems exist where the order of substrates in the chain elongation assembly pathway does not correspond with the order of genes on the BGC[29]. In these non-colinear gene assemblies, some of the substrates in the pathway differ from the order of genes by a series of inversions, exchanges, and insertions (Supplementary Fig. 4).

To address this, Seq2PKS recruits a pairwise nearest neighbor (PNN) search approach that utilizes the docking domain motifs at the two ends of each gene to predict the correct assembly pathway (Fig. 3). In addition, since the majority of adjacent genes in PK BGCs are contiguous, Seq2PKS further rewards orders that preserve co-linearity. Figure 4 illustrates the fivefold cross-validation accuracy of the PNN method compared with DDAP, which employs a support vector machine algorithm[27]. As a benchmark dataset, we used docking domain sequences from 90 BGCs reported in the DDAP paper. The ROC-AUC curve for the PNN method is shown in Supplementary Fig. 5.

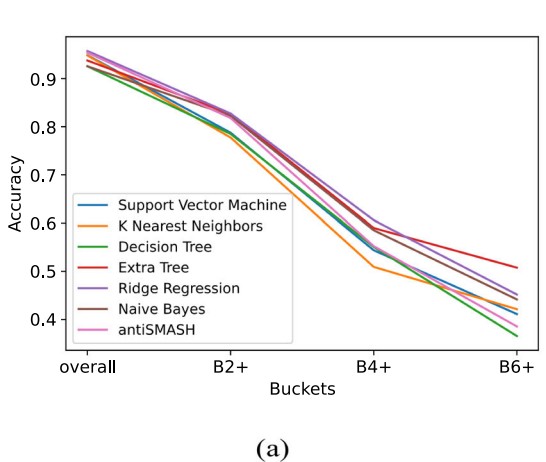

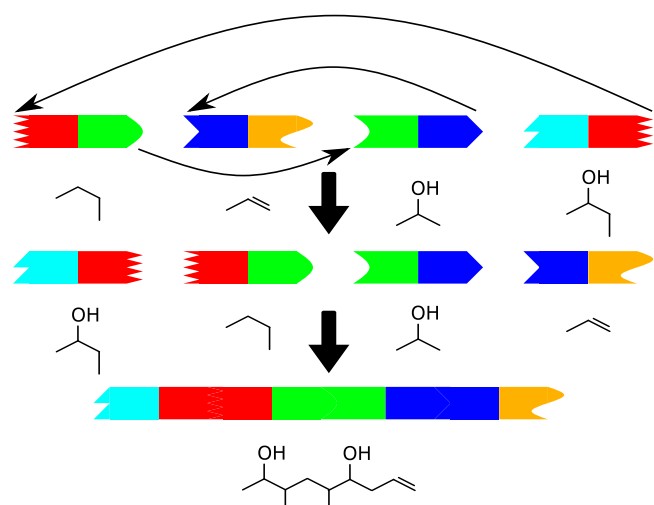

(a)

(b)

**Fig. 2 | The accuracy of different classifiers for predicting the substrate specificity of AT domains. a** The notation *Bk*+ represents the bin containing data points at least *k* Hamming distances away from any training data points. While most methods achieve similar accuracy, the extra-tree classifier generalizes better, i.e., it achieves higher accuracy for testing samples that are dissimilar to the training samples. **b** Confusion matrix for extra-tree prediction. The results are averaged across five different shuffles (used in fivefold cross-validation) and rounded to the nearest integer.

**Fig. 3 | TIMPKs assembly order pattern.** Many of the TIMPKs (Type 1 Modular Polyketides) are non-colinear, i.e., the order of assembly of substrates is not the same as the linear order of the modules on the PK genes. For non-colinear T1MPKs, we can identify the true order of substrates by identifying docking domains at the beginning (head) and end (tail) of modules that interact with each other. In the graph, the head and tail that are in the same color represent cognate docking domains.

In this comparison, PNN resulted in higher accuracies for both colinear and non-colinear BGCs. Also, PNN outperformed DDAP both with and without the feature of rewarding co-linearity.

To compare our method with antiSMASH's predictions for assembly order, we selected all 27 non-colinear PKSs from our benchmark dataset and predicted their assembly order using anti-SMASH. Concurrently, we extracted Seq2PKS interaction scores during our fivefold cross-validation for the testing samples using the best-performing model for Seq2PKS, thus ensuring no sample overlap between training and testing data. These interaction scores were then employed to rank the assembly orders. Of the 27 non-colinear samples, antiSMASH accurately predicted the assembly pathway for 20, while Seq2PKS correctly predicted 25 (Fig. 4).

We also tested the efficacy of PNN on 19 reported BGCs that are non-overlapping with our training data. Starting with the genome sequences containing these BGCs, Seq2PKS first identifies the genes and the docking domain sequences using HMM search before predicting the assembly order. For 14 out of 19 BGCs, Seq2PKS correctly identified the pathway genes. Among these 14, Seq2PKS correctly predicted the assembly order for 12 BGCs.

## Benchmarking post-assembly modification module

Seq2PKS predicts the mature structure of hypothetical polyketides by applying modifications corresponding to tailoring enzymes present in their BGCs. Thus, we created a database of known polyketide tailoring enzymes and their corresponding modifications by literature mining and parsed them in a computer-readable format. Our format consists of a motif (stored as a SMILES string) along with a series of graph modifications (addition/removal of nodes and edges) that are applied to the molecular structure if the corresponding motifs are observed (Fig. 5).

Seq2PKS first annotates all the modification enzymes in the BGC. Then, modifications corresponding to these enzymes are applied to the core structures foreseen in the previous step. Combined with the results of earlier steps, in 23 cases, Seq2PKS can correctly predict the mature molecular structure starting from the BGC, excluding the uncommon starter units (Table 1).

## Benchmarking against PRISM and antiSMASH

To better evaluate the power of Seq2PKS, we benchmarked the overall accuracy against the state-of-the-art methods PRISM[17] and antiSMASH (antiSMASH is not designed to predict the final compound structures)[24]. We obtained 80 known Cis-AT polyketide BGCs from MiBIG and calculated the Tanimoto similarity between the predicted mature and true structures. At a Tanimoto similarity threshold of 0.6, Seq2PKS found 13 molecules, while PRISM and antiSMASH identified two and zero molecules, respectively (Fig. 6). At a Tanimoto similarity threshold of 1, Seq2PKS found eight molecules, while PRISM and antiSMASH identified zero molecules. Since antiSMASH predicts only hypothetical core structures rather than mature polyketide compounds, a lower Tanimoto similarity between antiSMASH results and the actual compounds is expected. Moreover, Seq2PKS and PRISM predict multiple structures per BGC (to be refined by mass spectrometry in downstream steps), while antiSMASH predicts only one structure.

We further benchmarked versions of Seq2PKS that only consider cyclization modification or cyclization along with primary sugar

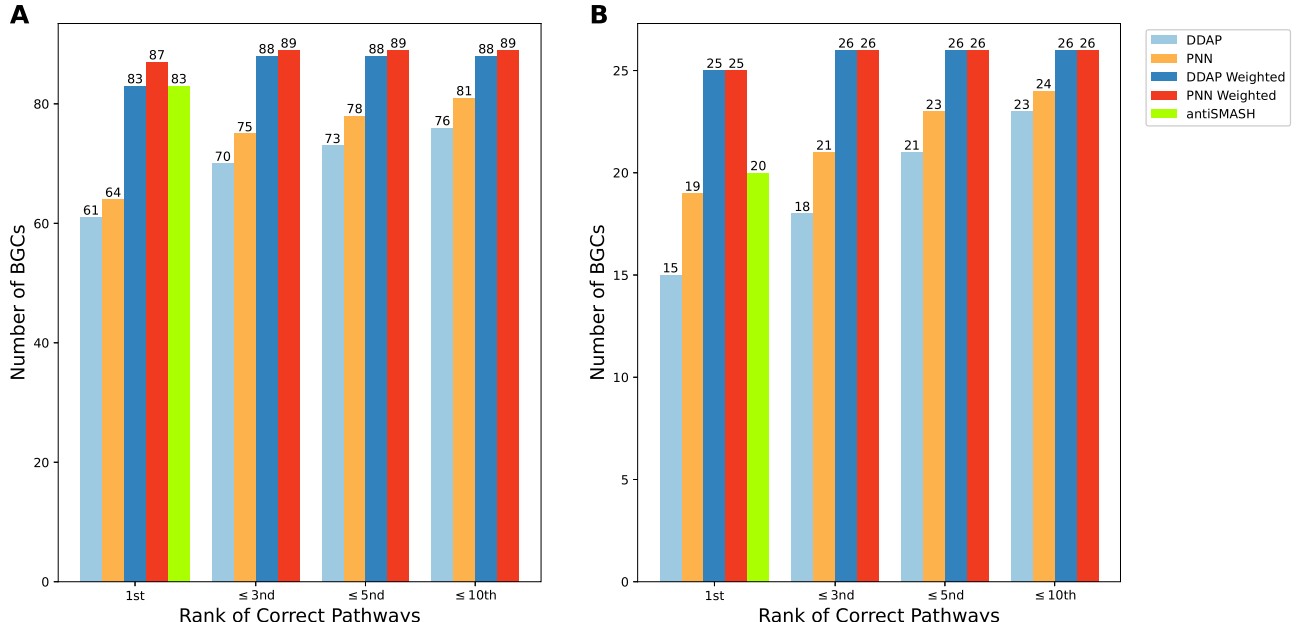

**Fig. 4 | Comparison of pathway prediction accuracy between PNN, antiSMASH, and DDAP.** The comparison is performed across (**A**) all 90 candidate BGCs and **B** 27 non-colinear BGCs in the dataset. In both scenarios, PNN outperforms DDAP and antiSMASH in accuracy. Note that for antiSMASH, only the rank one order is reported.

moieties (instead of all post-assembly modifications). Even these incomplete versions of Seq2PKS outperform PRISM[17]. At a Tanimoto similarity threshold of 0.6, the first version can identify seven molecules, and the second version can identify four molecules.

### Searching predicted structures against mass spectra
We further search against paired genomics and mass spectral dataset of *Streptomyces* strains available from GNPS using Dereplicator+ and Variable Dereplicator+[25]. As a result, five polyketides were identified, including nystatin (Supplementary Fig. 6), chalcomycin (Supplementary Fig. 7), salinomycin (Supplementary Fig. 8), aureothin (Supplementary Fig. 9), and vicenistatin (Supplementary Fig. 10). Nystatin, chalcomycin, and salinomycin do not have any uncommon starter unit, so their exact structures can be constructed. Therefore, they were identified through a Dereplicator+ search of hypothetical structures against mass spectra in exact mode. Aureothin and vicenistatin possess starter units, which the software cannot predict. Therefore, we relied on a modification-tolerant search utilizing Variable Dereplicator+ to identify them. Variable Dereplicator+ is a specialized version of Dereplicator+ that accommodates the absence of starter units by incorporating hypothetical mass shifts during the matching procedure.

### Identification of BGC for monazomycin
By taking the genome sequence of *Streptomyces* NRRL B12432 as input, Seq2PKS identified one long PK BGC and predicted 15 core structures leading to 4,035,840 mature compounds. Variable Dereplicator+ search of these structures against the corresponding spectra resulted in identifying a PK-spectrum match. Searching against the PubChem[30] database annotated this spectrum as monazomycin (Fig. 7). The predicted compounds correctly captured 23 out of 24 substrates involved in constructing the monazomycin core structure. The missing substrate is attributed to the absence of the corresponding module during the HMM search process. Furthermore, the accurate post-modification steps for compound construction were appropriately captured. Currently, there are no BGCs reported encoding for monazomycin biosynthesis, and we provide the plausible hypothetical biosynthetic assembly for this metabolite through our developed pipeline (Fig. 7). Experiments have demonstrated the effectiveness of monazomycin against gram-positive bacteria[31,32]. Our results enable access to the

monazomycin BGC, which unveils opportunities for pathway engineering to optimize production and generate analogs for structure-activity relationship studies.

### Identification of BGC for oasomycin A
The Seq2PKS application on the NRRL databases led to the identification of *Streptomyces* NRRL ISP-5512 genome with a putative long PK BGC. Seq2PKS generated 52 core structures, resulting in 9,845,760 mature compounds for this PK BGC. Searching the constructed mature compounds by spectral dataset using Variable Dereplicator+, we found one PK-spectrum match with a score of 46. We compared the matched spectrum against the PubChem[30] database and observed a high similarity to oasomycin A (Fig. 8). The predicted structure shares the identical core structure as oasomycin A with its unusual pattern of macrocyclization. Despite the initial failure to capture the cycloaddition modification during the compound construction process, it has been successfully identified through the variable search algorithm. Since no BGC has been reported for the biosynthesis of oasomycin A, we hypothesize that the identified BGC is a putative BGC for the construction of this fascinating natural product (Fig. 8).

### Experimental validation of the monazomycin BGC
To further corroborate the BGC identified for monazomycin, we extensively annotated the BGC (Supplementary Table 1). The annotation results confirmed the necessary biosynthetic machinery to produce monazomycin.

We cultivated *Streptomyces cinnamoneus* NRRL B-24434 and collected LC-MS/MS data on the butanolic extracts of its growth medium. Comparison with the authentic standard in both retention time (Supplementary Fig. 11) and tandem mass spectrometry data (Supplementary Fig. 12) was consistent with monazomycin production by *Streptomyces cinnamoneus* NRRL B-24434.

### Identification of actiphenol variant 2-aminobenzamide-actiphenol
We applied Seq2PKS to the *Streptomyces actiphen* genome available through the University of Michigan Natural Product Discovery Core and identified a long PK system with 55% similarity to the previously proposed actiphenol BGC. Seq2PKS generated one core structure,

**Fig. 5 | The list of PK modifications obtained in this study.** We constructed this database in a computer-readable format by mining the literature for known BGCs from the MIBiG database[52]. The modification sites are highlighted in red. In each case, the enzyme responsible for the modification is also shown.

resulting in 258 mature compounds for this PK BGC. By searching the constructed molecules in a spectral dataset using Dereplicator+, we identified two PK-spectral matches with scores of 21 and 28. We extensively annotated the BGC for these molecules (Supplementary Table 2) and compared it with the reported BGC for actiphenol from

MiBiG database (Supplementary Fig. 13). The former molecule is a previously reported variant of actiphenol called Nong-Kang 101-G (Supplementary Fig. 14)[33], while the latter is an actiphenol congener in which the cyclohexanone unit is substituted by a phenol moiety (Fig. 9)[34].

**Table 1 | Seq2PKS identifies the correct compound structure of 23 BGCs**

| Compound name | BGC ID | Mass | Starter unit | Correct pathway rank | # of Core structures | Best Tanimoto similarity |
|---|---|---|---|---|---|---|
| 4-Z-annimycin | BGC0001298 | 331.40 | False | 1/2 | 30 | 1.0 |
| Abyssomicin C | BGC0000001 | 346.40 | False | 1/6 | 16 | 1.0 |
| Chalcomycin A | BGC0000035 | 700.82 | False | 1/120 | 4 | 1.0 |
| Chlorothricin | BGC0000036 | 941.46 | False | 1/120 | 128 | 1.0 |
| Concanamycin | BGC0000040 | 692.90 | False | 1/720 | 16 | 1.0 |
| Halstoctacosanolide A | BGC0000073 | 845.12 | False | 1/5040 | 42 | 1.0 |
| Herboxidiene | BGC0001065 | 438.60 | False | 1/6 | 16 | 1.0 |
| Lasalocid | BGC0000086 | 612.80 | False | 1/5040 | 2 | 1.0 |
| Methymycin | BGC0000094 | 469.62 | False | 1/24 | 2 | 1.0 |
| Nystatin | BGC0000115 | 926.10 | False | 1/720 | 80 | 1.0 |
| Pimaricin | BGC0000125 | 665.70 | False | 4/120 | 32 | 1.0 |
| Pladienolide B | BGC0000126 | 536.70 | False | 1/24 | 15 | 1.0 |
| Salinomycin | BGC0000144 | 751.00 | False | 1/362,880 | 2 | 1.0 |
| Spinosad A | BGC0000148 | 731.97 | False | 1/120 | 15 | 1.0 |
| Streptoseomycin | BGC0001784 | 599.60 | False | 1/1 | 26 | 1.0 |
| Tetrocarcin A | BGC0000162 | 782.90 | False | 22/120 | 128 | 1.0 |
| Ansamitocin P-3 | BGC0000020 | 635.15 | True | 1/24 | 8 | 0.69 |
| Aureothin | BGC0000024 | 397.43 | True | 1/6 | 2 | 0.58 |
| Mycinamycin | BGC0000102 | 727.89 | True | 1/120 | 8 | 0.56 |
| Phenylnannolone A | BGC0000122 | 278.35 | True | 1/1 | 4 | 0.74 |
| Soraphen A | BGC0000147 | 520.70 | True | 1/2 | 2 | 0.46 |
| Spinosad A | BGC0000148 | 731.97 | True | 1/120 | 15 | 0.86 |
| Vicenistatin | BGC0000167 | 500.70 | True | 2/24 | 30 | 0.72 |

For compounds with uncommon starter unit, Seq2PKS can recover the compound structure except for the starter unit.
The column # of Core Structures shows how many theoretical core structures are being generated by Seq2PKS. The top compounds do not have starter units, while the bottom ones have.

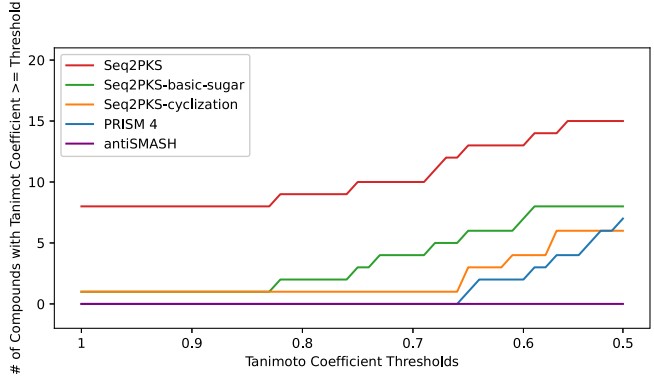

**Fig. 6 | Tanimoto similarity coefficient between the mature compounds predicted from 80 MiBIG BGCs.** Seq2PKS-cyclization and Seq2PKS-basic-sugar represent versions of Seq2PKS that only consider cyclization, or cyclization plus addition of basic sugars as post-translational modifications, respectively. The similarity is obtained from Skinnider et al. Note that Seq2PKS and PRISM generate multiple structures per BGC (which are further refined using mass spectrometry data), while antiSMASH generates a single core structure (solely based on genomics data).

The actiphenol molecule was isolated as a yellow solid with a molecular formula of $C_{22}H_{23}N_3O_5$, derived from an HRESIMS ion peak of $C_{22}H_{22}N_3O_5$ [M-H]$^-$ ($m/z$: found 408.1455, calcd 408.1559; NMR data for the molecule is shown in Supplementary Figs. 15, 16, 17, 18, 19, 20 and Supplementary Table 3). Careful analyses of the 1D and 2D NMR spectra confirmed the actiphenol moiety, however, the C-4 methyl group is absent. Instead, we observed a methylene ($\delta_H = 4.30$) at C-15 ($\delta_C = 45.9$) that correlates with ($\delta_C = 136.9$), C-5 ($\delta_C = 127.6$), and an aromatic carbon C-16 (chemical shift of ($\delta_C = 149.9$) atypical of the

actiphenol spin systems. Further examination revealed the presence of an additional aminobenzamide moiety attached to the actiphenol through an N-C bond. The HMBC correlations between the triplet H-18 ($\delta_H = 7.23$)/C-16 ($\delta_C = 149.9$) and the doublet H-20 ($\delta_H = 7.61$)/C-16 ($\delta_C = 149.9$) and C-22 ($\delta_C = 171.9$) confirmed the relative position of the secondary amine ($\delta_H = 7.23$) and the primary amide on this aromatic ring. Based on its structure, we named this molecule 2-aminobenzamide-actiphenol (Supplementary Fig. 21).

We tested the molecules against two cancer cell lines, SW-48 (CCL-231, colon cancer) and HCT15 (CCL-225 colon cancer), based on the initial strong cytotoxic activity observed (Supplementary Fig. 22) from the crude extracts generated by the producing strain (*Streptomyces actiphen*) during the initial isolation. However, the isolated molecule 2-aminobenzamide-actiphenol did not show appreciable activity upon isolation and characterization (Supplementary Fig. 23).

## Discussion

This study presents Seq2PKS, a machine learning approach aimed at enhancing the process of polyketide discovery. Seq2PKS utilizes large-scale mass spectral and genomic datasets generated from diverse microbial isolates. Seq2PKS addresses various complexities involved in predicting and characterizing polyketide structures from biosynthetic gene clusters. By constructing an extensive tailoring modification database and applying it to the cyclized core structure, Seq2PKS achieves the correct mature structure for 16 molecules. In the case of polyketides with uncommon starter units, Seq2PKS successfully recovers the compound structures for seven molecules, with the exception of the starter units.

In addition to genomic-driven structural predictions, Seq2PKS is also an automated method that integrates the power of metabolomics (mass spectrometry) with the genomics data for scalable polyketide identification. Searching the paired genomics and metabolomics

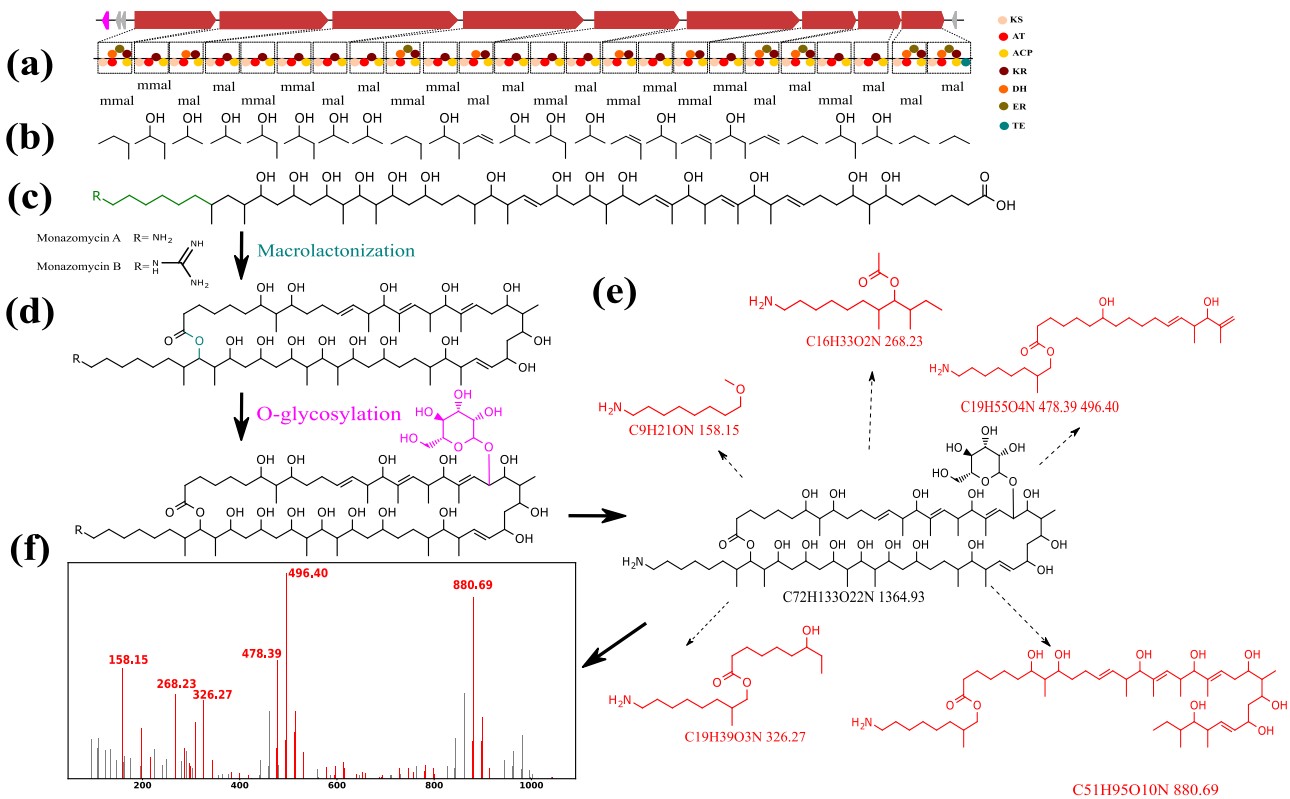

**Fig. 7 | Identification of monazomycin BGC by Seq2PKS. a** Domains and modules in the biosynthetic genes are identified. **b** Substrate specificity for each domain is predicted. **c** Assembly order is predicted, and the core structure is constructed by using the predicted substrates for each module. **d** Corresponding modifications are applied to the core structure to form hypothetical molecules. **e** Annotation of monazomycin's spectrum using the Dereplicator+ model. Fragments of monazomycin matching a peak in the spectrum are highlighted. **f** The annotated spectrum of monazonmycin with peaks matching fragment masses is shown in red.

dataset of *Streptomyces* strains with this feature, Seq2PKS correctly identified three known polyketides that lacked starter units. Two additional known polyketides with starter units were identified using Dereplicator+ in variable search mode on the Seq2PKS outputs. In addition, Seq2PKS identified potential BGCs for two polyketides, monazomycin and oasomycin A, whose BGCs were unknown. Further, Seq2PKS identified an actiphenol variant 2-aminobenzamide-actiphenol that was missed by previous bioactivity-driven efforts due to the lack of cytotoxic activity.

Polyketides, a significant class of natural products with numerous compounds in clinical use, have a myriad of therapeutic applications. However, accurately predicting their chemical structures can be challenging due to the intricacies of polyketide biosynthesis and the occurrence of complex modifications and cyclization patterns. Seq2PKS offers several innovative enhancements over the existing methodologies. First, it utilizes an extra-tree-based classification algorithm for domain specificity prediction, which improves the classification of domains that are not closely related to known ones. This approach increases the accuracy of substrate specificity predictions based on AT domain signatures.

Moreover, Seq2PKS introduces an innovative approach for predicting the assembly pathway. By considering the gene order in the genome, Seq2PKS accounts for the similarities between substrate order in polyketides and gene order in their BGCs, enabling more precise predictions of the substrate order. Also, by calculating the probabilities instead of predictions, Seq2PKS can evaluate multiple candidates from the output result.

Post-assembly modifications constitute a crucial step in polyketide biosynthesis that transforms the polyketide core structure into mature, bioactive molecules. Existing methods, such as antiSMASH[24], are designed to predict the core structures, rather than the final

compounds, of various natural products. While PRISM[17] incorporates post-modifications into its analysis, its focus is primarily on different cyclizations, overlooking other complex modifications. In contrast, Seq2PKS predicts multiple mature polyketide compounds by considering a range of potential modifications and refines these predictions with mass spectrometry data. To do this, Seq2PKS relies on a curated dataset of 78 enzyme-enabled modifications derived from the literature. These modifications are stored as SMILES strings and graph modifications and enable Seq2PKS to construct hypothetical structures of mature polyketides by applying diverse combinations of identified accessory enzyme modifications. This holistic approach broadens the chemical space explored and improves the accuracy of predicted polyketide structures.

Compared to other methods, in AT domain specificity prediction, Seq2PKS achieves over 94% accuracy, comparable to the accuracy for antiSMASH. However, Seq2PKS improves prediction accuracy for AT domain specificity by using an extra-tree-based model optimized to achieve high accuracy for test data points that are far from training data. For testing samples that are at least six Hamming distances away from any training data points, Seq2PKS achieves an accuracy of 51%, compared to 38% for antiSMASH. During the assembly line order prediction process, Seq2PKS employs a logistic regression algorithm, taking the docking domain sequences as input to predict the optimal assembly pathway for candidate core structures. This method achieves 92% accuracy for non-colinear polyketide BGCs, while DDAP and antiSMASH achieve 55% and 74% accuracy, respectively. Furthermore, Seq2PKS uses a subgraph isomorphism algorithm to consider a more comprehensive set of post-assembly modifications. Overall, Seq2PKS correctly predicts eight out of 80 polyketides in the benchmark dataset (16 if counting polyketides without PRISM result), while PRISM and antiSMASH do not predict any of the polyketides correctly. The

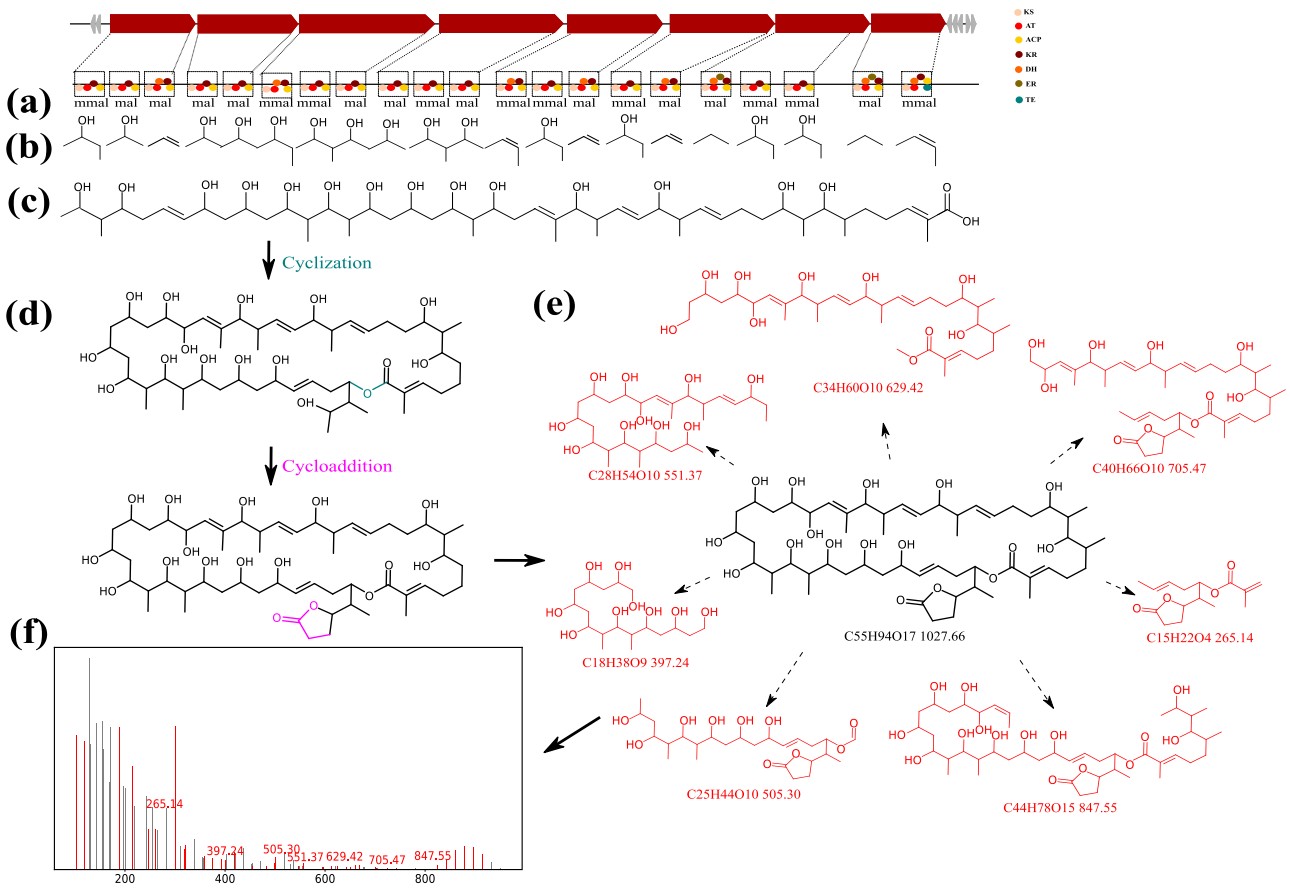

**Fig. 8 | Identification of oasomycin A BGC by Seq2PKS. a** Domains and modules in the biosynthetic genes are identified. **b** Substrate specificity for each domain is predicted. **c** Assembly order is predicted, and the core structure is constructed by using the predicted substrates for each module. **d** Corresponding modifications are applied to the core structure to form hypothetical molecules. **e** Annotation of oasomycin A spectrum using the Dereplicator+ model. Fragments of oasomycin A matching a peak in the spectrum are shown in red (**f**). **f** The annotated spectrum of oasomycin A with matched peaks is shown in red.

highest Tanimoto similarity that PRISM and antiSMASH achieve are 0.65 and 0.30.

Seq2PKS integrates mass spectrometry data into the polyketide discovery process. Mass spectrometry is a powerful tool for analyzing metabolites produced by organisms. Currently, two types of tools bridge metabolomics and genomic data. Correlation-based approaches like NPlinker[35], NPOmix[36], and other metabolomics methods[37–39] focus on linking gene cluster families (GCFs) with molecular families (MFs) or spectra, based on co-occurrence of molecular features and BGCs. Conversely, feature-based approaches like GNP[18], MetaMiner[40], Seq2RiPP[41], and NRPminer[42] strive to associate BGCs with mass spectra by predicting the hypothetical structure of BGC products, followed by in silico mass spectral database search. Seq2PKS is a feature-based method that integrates metabolomics and genomics data capabilities for the efficient and scalable identification of polyketides. By integrating mass spectrometry data, Seq2PKS can identify polyketides with unknown modifications and validate their presence in microbial samples. Seq2PKS produces multiple molecular structures per BGC and ranks them against paired mass spectrometry data, resulting in increased accuracy and reliability of polyketide structure predictions.

Currently, there is no accurate method for predicting structures of starter units. In contrast to other polyketide substrates, starter unit structures are very diverse and can even be unique to specific polyketides and their homologs. This makes it infeasible to design machine learning models for accurately predicting the starter unit of polyketides without overfitting to training data. To overcome this challenge, Seq2PKS relies on variable mass spectral database search methods that can identify the overall mass of the starter unit.

Actiphenol, a prominent member of the glutarimide-containing polyketide family, has long been recognized for its role as a eukaryotic translation inhibitor. Seq2PKS unveiled a variant of actiphenol, named 2-aminobenzamide-actiphenol, from *Streptomyces actiphen*. This molecule diverges notably from classical actiphenol structures by lacking a methyl group at C-4 and introducing both a methylene group and an aminobenzamide moiety. Although initial tests revealed promising cytotoxic effects in crude extracts against colon cancer cell lines, the isolated variant did not exhibit substantial activity in further assays. Further studies are needed to explore the bioactive properties of this compound.

The insights obtained from this study provide a foundation for further exploration and advancement in the field of polyketide discovery. We showed that Seq2PKS is a powerful tool for exploring the immense potential of polyketides as sources of drugs. The ongoing integration of advanced computational techniques, machine learning, genomics, and mass spectrometry data enable continued innovations in the discovery and development of polyketide-based therapeutics.

Despite the promising advancements brought about by Seq2PKS, it faces limitations that warrant further attention. Firstly, while the curated dataset of modifications is comprehensive, it is not exhaustive. There may still be unknown or uncharacterized modifications in polyketide biosynthesis that are not accounted for in the current version of Seq2PKS. Secondly, the current approach only focuses on

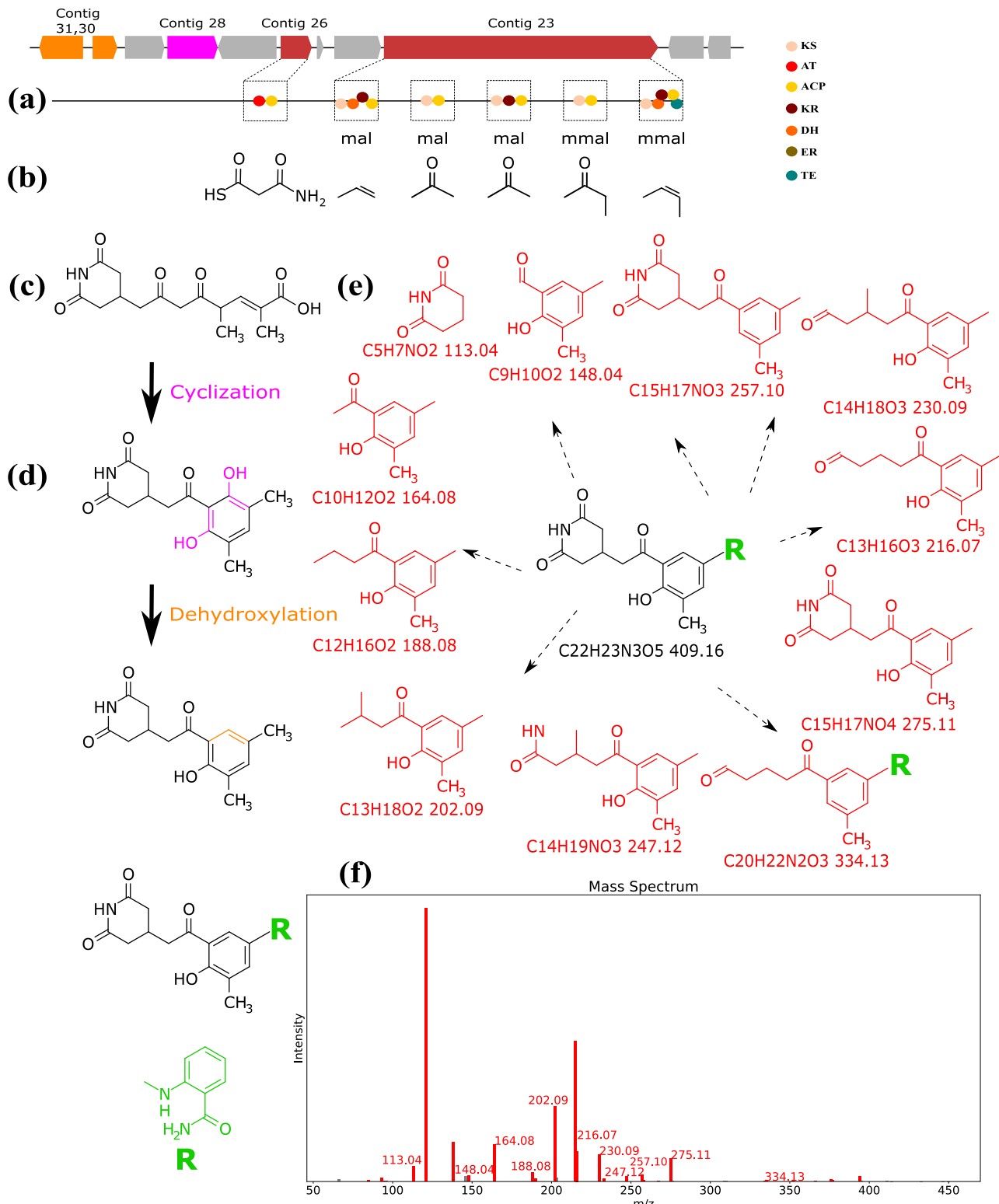

**Fig. 9 | Identification of 2-aminobenzamide-actiphenol by Seq2PKS. a** Domains and modules in the biosynthetic genes are identified. **b** Substrate specificity and mature substrate for each module are predicted. **c** Assembly order is predicted, and the core structure is constructed by connecting the predicted mature substrates for each module. **d** Cyclization and dehydroxylation modifications are applied by two enzymes from the BGCs (shown in pink and orange, respectively). While Seq2PKS cannot predict the presence of 2-aminobenzamide, it correctly predicts its molecular formula from the difference between the precursor mass spectrum and the total mass of the hypothetical molecule. **e** Fragments from the hypothetical molecule (shown in black) that match a peak in mass spectra are highlighted (shown in red). Fragments are generated by one or two rounds of fragmentation of the hypothetical molecule. **f** The annotated spectrum of the hypothetical molecule is shown in red.

Cis-AT polyketides and might not accurately predict the structures of other types of polyketides. Lastly, Seq2PKS pipeline relies heavily on mass spectrometry and is limited by the availability and quality of this data. These limitations highlight areas for potential improvement and provide avenues for future research in the field of polyketide discovery.

## Methods

Seq2PKS mines the microbial genomes to identify BGCs, and predicts millions of hypothetical molecular structures per BGC. Afterward, these structures are searched against mass spectral data collected from the extracts of microbial cultures using Dereplicator+[25]. These steps are described below:

(a) **Annotating PK domains and enzymes**. The polyketide BGCs are identified using antiSMASH[24]. Given a genome (or a set of contigs), antiSMASH uses HMMER[43] to find enzymes that are specific to PK gene clusters. The portion of the genome within 10,000 bp upstream and downstream of these enzymes are defined as polyketide BGCs. Seq2PKS detects domains and tailoring enzymes in polyketide BGCs by searching for their hidden Markov model profiles[43].

(b) **Predicting substrates**. Previous studies have shown that the substrate specificity of AT domains is largely determined by specific amino acid residues of the AT active site pocket[44]. Therefore, for each AT domain, Seq2PKS extracts 24 active site residues reported by Yadav et al.[19] by aligning them to the reference sequences using MUSCLE sequence alignment[28]. These 24 active site residues were identified by analyzing the crystal structure of E. coli FAS AT domain[45] and have proven to show distinct patterns across different AT domain specificities. Seq2PKS represents these signatures as one-hot encoded vectors, where each amino acid maps to a unique twenty-dimensional binary vector containing nineteen zeros and a single one. An extra-tree-based model is trained to predict the domain specificity based on these vector representations. The number of trees in the forest is set to 150 and the max depth of the tree is set to 10 to achieve optimal performance.

Following the initial substrate identification from an AT domain, other domains within the same module, including ketoreductase (KR), dehydratase (DH), and enoylreductase (ER) domains, can modify the substrate further. Specifically, the KR domain catalyzes the reduction of the substrate carbonyl group to a hydroxy group. A DH domain then removes a water molecule from the hydroxy substructure, leading to the formation of an olefin substructure. Finally, an ER domain reduces the double bond introduced by the DH domain, resulting in saturation of the polyketide chain. This sequence of modifications contributes to the remarkable structural diversity and complexity of polyketide compounds. Based on this logic, Seq2PKS recruits a rule-based approach (Supplementary Fig. 3) that predicts each substructure, i.e., modified substrate, using the substrate specificity of the AT domain and the presence of substrate-modifying domains in the module).

(c) **Predicting the assembly pathway**. A single helical segment from the C-terminal linker of one MT1PKS open reading frame (ORF) and three helical segments from the N-terminal linker of the subsequent MT1PKS ORF represent structures known as docking domains which direct the interactions of neighboring MT1PKS proteins in the assembly pathway[46]. An ORF can follow another ORF in the assembly line only if the docking domain at the C-terminus of the former MT1PKS ORF can interact with the N-terminal docking domain of the latter MT1PKS ORF. Therefore, interactions between docking domains can be used to infer the order of ORFs in the assembly line. Seq2PKS calculates a docking interaction score between terminal modules in a BGC. Each

module's head (i.e., N-terminal) and tail (i.e., C-terminal) docking domains are extracted as the first 100 amino acid residues from the C-terminus and the first 50 amino acid residues from the N-terminus, respectively. The head domain from the first module and the tail domain from the last module of the MT1PKS pathway are excluded.

For each head (tail) docking domain in the training set, Seq2PKS calculates the BLAST bit-score between that head (tail) docking domain and every head (tail) docking domain. The distance between any two head–tail pairs is defined as the sum of the bit-score between heads and between tails.

For each head–tail pair in the training set, $n$ top-scored interacting training pairs with the largest scores (denoted as interacting neighbors), and the $n$ top-scored non-interacting training pairs (denoted as non-interacting neighbors) were selected ($n = 3$). Then, for each interacting/non-interacting neighbor, the bit-score between the neighboring heads and the input head, and the bit-score between the neighboring tails and the input tail were used as features, resulting in $2n$ features. We used 382 pairs of head and tail sequences from interacting (consecutive) modules as positive training data points, and the remaining 1622 pairs from non-interacting modules in the same BGC were used as negative samples. 382 negative samples were randomly selected from all the negative samples to balance the training set. These training samples were used to train a logistic regression model that predicts whether a pair is interacting.

Given a BGC, first the interaction scores between each pair of head and tail from different genes in the BGC are calculated using the trained model. Then, the candidate pathways are generated by calculating all possible permutations of genes in the BGC, and the overall interaction score for each pathway is calculated as follows:

$$S_{pathway} = \sum_{p \in Non-adj} S_p + \sum_{p \in Forward} S_p.W_{forward} + \sum_{p \in Backward} S_p.W_{backward}$$

(1)

Where *Non-adj*, *Forward*, and *Backward* represent the set of gene pairs in the pathway that are not adjacent, in forward order, and in backward order in the BGC, respectively, $S_p$ is the interaction score of pair $p$. Parameters $W_{forward}$ and $W_{backward}$ are set to one by default. The candidate pathways are then ranked by their overall interaction scores (Supplementary Fig. 24).

(d) **Incorporating post-assembly modifications**. Once the polyketide core structures are synthesized, they undergo various post-assembly modifications. These modifications are chemical alterations driven by tailoring enzymes that affect the properties and functionalities of the polyketides. Key modifications include the addition or removal of phosphate groups (phosphorylation and dephosphorylation)[47], the attachment of acetyl groups (acetylation)[48], and the incorporation of sugar molecules (glycosylation)[49]. Seq2PKS predicts hypothetical structures of mature polyketides by applying various combinations of modifications corresponding to the identified tailoring enzymes. To this end, we extracted polyketide tailoring enzymes and their corresponding modifications by literature mining and parsed them in a computer-readable format (Supplementary Fig. 25). For each modification, the reaction motif is stored as a SMILES string, along with a series of graph modifications (addition/removal of nodes/edges) that are applied to the motif.

Given a polyketide BGC, Seq2PKS extracts all the tailoring enzymes in the BGC using HMMER. For each enzyme, it searches for the corresponding motif in the core polyketides using the Ullman algorithm[50]. Afterward, for each core structure, all the

combinations of possible reactions are applied to the matched motifs to generate candidate products. This approach generates a total of $k \times (m_1 + 1) \times (m_2 + 1) \times \ldots \times (m_n + 1)$ hypothetical mature polyketides, where $k$ is the number of core polyketides, $n$ is the number tailoring enzymes, $m_n$ is the number of motifs present in the core polyketide for each enzyme.

(e) **Searching mature structures against mass spectra using Dereplicator+ and Variable Dereplicator+.** Seq2PKS searches the hypothetical polyketide structures predicted in the previous steps against mass spectral data using the exact search method Dereplicator+ and variable search method Variable Dereplicator+[25]. Using variable spectral search with Variable Dereplicator+, Seq2PKS can identify polyketides with unknown substrates and modifications.

## Cross-validation for machine-learning model

During the substrate specificity prediction and the assembly line order prediction process, standard machine learning models are employed. These models include Logistic Regression, Support Vector Machine, K Nearest Neighbor, Multilayer Perceptron, Random Forest, Decision Tree, Bernoulli Naive Bayes, Gaussian Naive Bayes, and Extremely Randomized Trees. To ensure there is no sample overlap, fivefold cross-validation is utilized, where the datasets are randomly split into five subsets with the same number of samples. In each step, four of the subsets are used for training, and the remaining one serves as test data.

Fivefold cross-validation offers a more accurate estimation of model performance, particularly when the training data is limited. This process is executed five times for each selected model, with the dataset being randomly shuffled each time. The accuracies are calculated by averaging across these cross-validations.

## Validation of monazomycin BGC

We cultivated liquid cultures of *Streptomyces cinnamoneus* NRRL B-24434 in ISP2 medium for 7 days at 28 °C. The shaking speed is 220 rpm with a flask size of 1 L and medium content of 100 mL. This was followed by extraction of the broth with 1-butanol, rotary evaporation of the 1-butanol extract, resuspension of the extract in 80% methanol, followed by LC-MS/MS analysis. LC-MS data was collected on a Thermo QExactive Orbitrap connected to a Vanquish LC system. LC settings were as follows: injection volume 5 µL; Phenomenex Kinetex 2.6 µm C$_{18}$ reverse-phase 100 Å 150 mm × 3 mm LC column; LC gradient, solvent A, 0.1% formic acid; solvent B, acetonitrile (0.1% formic acid); 0 min, 10% B; 5 min, 60% B; 5.1 min, 95% B; 6 min, 95% B; 6.1 min, 10% B; 9.9 min, 10% B; 0.5 mL/min. MS settings were as follows: positive ion mode; full MS, resolution 70,000; mass range 400–1200 $m/z$; dd-MS2 (data-dependent MS/MS), resolution 17,500; AGC target $1 \times 10^5$, loop count 5, isolation width 1.0 $m/z$, collision energy 25 eV, dynamic exclusion 0.5 s.

## Genome extraction, sequencing, assembly, and annotation

The genomic DNA for 44321-A2 was extracted using BIOSEARCH Technologies MasterPure Complete DNA and RNA purification kit following the manufacturer's protocol with some modifications, including additional lysis steps using EDTA (50 mM) and lysozyme (10 mg/mL) and one more step of heat treatment at 95 °C before RNase A treatment. The genomic DNA was sequenced at the University of Minnesota, Genomics Center, Minneapolis, MN. The sample library was prepared using PacBio Sequel II HiFi—the SMRT Cell 8M typically generates ~4–5 million raw reads with flexible sequencing run times of up to 30 h, yielding a 1.04 GB fastq file. Demultiplexing and quality control were done using PATRIC service which obtained a total of 90,490 read pairs. The 44321-A2 was assembled using Canu version 1.7.1 and 2 rounds of polishing done as iteration using Racon version 2.4.13 as a part of the comprehensive genome analysis service at PATRIC. QUAST version 5.0.2, minimap2 (2.17-r974-dirty), samtools

Version 1.11, and Bandage 0.8.1 with default parameters were used for assembly quality assessment and visualization. The resulting assembled genome has an estimated length of 8,799,402 bp and an average GC content of 72.18%. Two contigs generated for this genome have 7647 protein-coding sequences (CDS), 68 transfer RNA (tRNA) genes, and 18 ribosomal RNA (rRNA) genes. The annotation included 2918 hypothetical proteins and 4729 proteins with functional assignments. The proteins with functional assignments included 1270 proteins with Enzyme Commission (EC) numbers, 1111 with Gene Ontology (GO) assignments, and 1007 proteins that were mapped to KEGG pathways. Then the assembled contig FASTA file is used to extract the 16S fragments with the help of ContEst16S tool at Ezbiocloud. Whole-genome similarity metrics, including average nucleotide identity (ANI) and DNA-DNA hybridization (DDH), were obtained to estimate genetic relatedness and define phylogeny. FastANI showed 86% genome-relatedness of our microbial strain genome to *Streptomyces atratus* ASM333086v1 and *Streptomyces gelaticus* ASM1464953v1. A whole-genome-sequence-based phylogenetic tree was built using the TYGS analysis method further supporting the genetic closeness between 44321-A2 and the *Streptomyces* as mentioned above. The TYGS database confirmed that 44321-A2 might potentially be an unknown species and, therefore, named the strain *Streptomyces actiphen* based on its ability to produce varied actiphenol analogs.

## General NMR and LC-HRMS/MS materials and methods

Nuclear magnetic resonance (NMR) spectra were acquired utilizing either a Bruker 600 NMR spectrometer ($^1$H: 600 MHz, $^{13}$C: 150 MHz) featuring a Magnex 600/54 active shielded premium magnet, a Bruker liquid N2 cooled Prodigy cryoprobe, and a Bruker NEO600 console, or a Bruker 800 NMR ($^1$H: 800 MHz, $^{13}$C: 200 MHz) equipped with an Ascend magnet with active shield, a 5 mm triple-resonance inverse detection TCI cryoprobe, and a Bruker NEO console. MestReNova NMR software was employed for all NMR data analyses. Residual solvent peaks were used as references for chemical shift values [$^1$H (DMSO-d6): 2.50 ppm; $^{13}$C (DMSO-d6): 39.51 ppm].

LC-HRMS/MS analyses of Biotage fractions, HPLC fractions, and purified compounds were conducted using an Agilent 1290 Infinity II UPLC coupled to an Agilent 6545 ESIQ-TOF-MS system operating in both positive and negative modes. A Phenomenex Kinetex 1.7 µ Phenyl-Hexyl 100 Å (2.1 × 50 mm) column was utilized for chromatography, with a 2 µL injection volume per sample. Samples were eluted using a gradient starting with a 1 min isocratic wash step consisting of 90% A (95% H$_2$O/5% MeCN with 0.1% formic acid) and 10% B (100% MeCN with 0.1% formic acid), followed by a 6 min linear gradient step from 10% B to 100% B, and ending with 2 min of 100% B wash at a flow rate of 0.4 mL/min. The divert valve was set to MS for 0–7.4 min and to waste from 7.4 to 9 min. The dual AJS ESI conditions were: gas temperature at 320 °C, sheath gas temperature at 350 °C, sheath gas flow rate at 11 L/min, and source capillary voltage at 3500 V. The mass range of MS was set to 100–2000 $m/z$, with an acquisition rate of 10 spectra per second. The mass range of MS/MS was set to 50–2000 $m/z$, with an acquisition rate of 6 spectra per second and an isolation width of ~1.3 $m/z$. The collision energy was calculated using the formula: collision energy = $(5 \times m/z)/100 + 10$. The maximum precursor per cycle was set to 9, and the MS/MS mass error tolerance was ± 20 ppm. Reference masses for positive mode were purine C$_{18}$H$_{18}$F$_{24}$N$_3$O$_6$P$_3$ [M + H]$^+$ ion ($m/z$ 121.050873) and hexakis(1H,1H,3H-tetrafluoropropoxy)phosphazine C$_{18}$H$_{18}$F$_{24}$N$_3$O$_6$P$_3$ [M + H]$^+$ ion ($m/z$ 922.009798), while for negative mode, trifluoroacetic acid (TFA) C$_2$HF$_3$O$_2$ [M − H]$^-$ ($m/z$ 112.985587) and hexakis(1H,1H,3H-tetrafluoropropoxy)phosphazine C$_{18}$H$_{18}$F$_{24}$N$_3$O$_6$P$_3$ [M + TFA − H]$^-$ ($m/z$ 1033.988109) were used. ACS grade solvents were used for Biotage fractionation, and HPLC grade or better solvents were used for HPLC purification and LC-HRMS/MS analyses, unless otherwise stated. All LC-MS/MS chromatograms, extracted base peak chromatograms (BPCs), and UV traces at 254nm

were subtracted from the chromatograms of the methanol (MeOH) blank. GraphPad Prism version 9.4.1 for Mac OS X (GraphPad Software) was used for data visualization and plotting.

## Fermentation of *Streptomyces actiphen*

*Streptomyces actiphen* 44321-A2 was streaked onto R2YE agar containing 5 g of yeast extract, 103 g of sucrose, 10 g of dextrose, 0.1 g of casamino acid, 0.25 g of $K_2SO_4$, 10.12 g of $MgCl_2 \cdot 6\,H_2O$, 5.73 g of TES buffer, 2 mL of trace element solution (containing 10 mg of $(NH_4)_6Mo_7O_24 \cdot 4\,H_2O$, 10 mg of $Na_2B_4O_7 \cdot 10\,H_2O$, 10 mg of $MnC_l2 \cdot 4\,H_2O$, 10 mg of $CuCl_2 \cdot 2\,H_2O$, 200 mg of $FeCl_3 \cdot 6\,H_2O$, 40 mg of $ZnCl_2$, and 1 L of deionized water, filter sterilized), 10 mL of 0.5% $KH_2PO_4$, 4 mL of 5 M $CaCl_2 \cdot 2\,H_2O$, 5 mL of 20% L-proline, 7 mL of 1 N NaOH, 25 µg/mL nalidixic acid, 10 µg/mL benomyl, 15 g of agar, and 1 L of double-distilled water. Plates were incubated for 5–7 days at 28 °C. For the strain, 3 mL seed cultures in 14-mL dual-position cap tubes were inoculated with a loopful of vegetative cells from R2YE plates and incubated for 5 days at 28 °C, 200 rpm; 3 mL seed cultures were then inoculated into 100 mL seed cultures in 250 mL baffled flasks and incubated for 7 days at 28 °C, 200 rpm; 50 mL of seed cultures were inoculated into 1 L of fermentation media in 2.8 L baffled Fernbach flasks and grown for 7 days at 28 °C, 200 rpm.

On day 7 of the fermentation, 25 g of Amberlite XAD16 resin contained within a polypropylene mesh bag was added to each fermentation culture and agitated overnight at 28 °C, 200 rpm. On day 8, all resin bags were removed and thoroughly washed with deionized water to remove any water-soluble media components and residual cell mass adsorbed on the resin bags. Each washed resin bag was extracted with 250 mL of methanol (MeOH) and 250 mL of ethyl acetate (EtOAc). The combined organic fractions were dried under vacuum and redissolved in a minimal amount of MeOH. The solutions were then centrifuged, and the supernatants were loaded onto C18 resin and dried under vacuum prior to Biotage C18 fractionation.

## Purification of 2-aminobenzamide-actiphenol

Preparative RF-HPLC fractionation was performed using the same Shimadzu LC-20AP system equipped with a reverse-phase Phenomenex Kinetex® 5 µm C18 100 Å (250 mm × 21.2 mm) column. The materials of strain *Streptomyces actiphen* (brought up in methanol at ~100–200 mg/mL) were eluted with a flow rate of 20 mL/min and a linear gradient starting with a 2 min isocratic wash step using 10% acetonitrile/$H_2O$ (with 0.01% TFA), then a 30 min linear gradient step from 10% acetonitrile/$H_2O$ (with 0.01% TFA) to 100% acetonitrile/$H_2O$ (with 0.01% TFA), and then a 5 min wash with 100% acetonitrile/$H_2O$ (with 0.01% TFA) followed by an 8 min equilibration with 10% acetonitrile/$H_2O$ (with 0.01% TFA). Less than 0.05 mg of the compound is isolated per liter of broth.

Preparative HPLC (Prep HPLC) HF16-18 was fractionated using 40 minutes of an isocratic method with 33% acetonitrile solvent B in $H_2O$ (both solvents A/B with 0.01% TFA) elution. The following compounds were purified using the same semi-preparative system with a reverse-phase Phenomenex Luna® 5 µm Phenylhexyl 100 Å (250 mm × 10 mm) column. 2-Aminobenzamide-actiphenol (~0.05 mg/L) was purified from HF16-18_F24-28 using 30 min of an isocratic 35% acetonitrile/$H_2O$ (with 0.01% TFA) elution.

## Compound entries

2-Aminobenzamide-actiphenol: yellow solid UV (MeOH) $\lambda_{max}$ (log $\varepsilon$): 218 nm (8.9), 267 nm (6.1), 355 nm (3.35); Mp: 197–201 °C; [^1]H NMR (600 MHz, *DMSO-d$_6$*): $\delta$ 2.17 (s), 2.39 (d, *J* = 16.5, 10.7 Hz), 2.58 (d, *J* = 16.5, 10.7 Hz), 2.58 (dd, *J* = 16.5, 10.7 Hz), 10.76 (s), 3.18 (d, *J* = 6.4 Hz), 3.18 (d, *J* = 6.4 Hz), 2.62 (m), 2.39 (dd, *J* = 16.5, 10.7 Hz), 4.30 (s), 4.30 (s), 8.47 (br s), 6.54 (t, *J* = 7.5 Hz), 6.69 (d, *J* = 8.4 Hz), 7.17 (br s), 7.23 (t, *J* = 7.9 Hz), 7.45 (s), 7.61 (d, *J* = 7.8 Hz), 7.82 (s), 7.86 (br s), 12.33 (s) [^13]C NMR (150 MHz, *DMSO-d$_6$*): $\delta$ 112.1 (CH), 114.9 (C), 114.9 (CH), 118.7 (C), 127.0

(C), 127.6 (CH), 129.5 (CH), 130.0 (C), 132.8 (CH), 136.9 (CH), 15.8 ($CH_3$), 171.9 (C), 173.4 (C), 205.6 (C), 37.5 ($CH_2$), 37.5 ($CH_2$), 37.5 ($CH_2$), 173.4 (C), 42.9 ($CH_2$), 42.9 ($CH_2$), 26.6 (CH), 37.5 ($CH_2$), 45.9 ($CH_2$), 45.9 ($CH_2$), 149.9 (C) HRESIMS: Ion peak of $C_{22}H_{22}N_3O_5$ [M-H]$^-$ *m/z*: found 408.1455, calcd 408.1559 (Supplementary Table 3).

## Cell-based bioactivity analysis

HCT15 (CCl-225) and SW-48 (CCL-231) cells were purchased from ATCC. All cell lines were Mycoplasma free and independently authenticated by short tandem repeat profiling, performed by ATCC. Cells were grown and cultured according to ATCC recommendations. HCT15 cells were cultured in RPMI1640 (30-2001) supplemented with 10% FBS (30-2020). SW-48 cells were cultured in Leibovitz's L-15 medium (30-2008) containing 10% FBS. SW-48 cells were grown and treated in an incubator set for atmospheric conditions (no supplemental $CO_2$ addition). For cell-based assays, cells were expanded and frozen into single-use aliquots. For each assay, cells were thawed at 37 °C for 1 min and then immediately resuspended in 10 mL of complete growth medium. Cells were then spun down at 300×*g* for 5 min and then resuspended in cell-specific growth medium and plated at 2500 cells per well into Greiner 781080 white cell culture 384-well plates with a total volume per well at 40 µL. Natural product extracts or fractions were dissolved in DMSO at 15 mg/mL and delivered into the assay plates using Echo 655 acoustic liquid handler instrumentation (Beckman Coulter). Extract and fraction testing concentrations were at 0.25%. For primary screening assays, extract testing was performed *n* = 1 at 0.25% final extract testing concentration (where the original fraction is defined at 100%). Validation assay and fraction studies were performed in triplicate at similar testing concentrations. Negative controls (medium only plus matching 0.25% DMSO) were included in columns 1 and 2. The positive control for these studies was a 10 µM treatment with staurosporine in columns 23 and 24 of each assay plate. Samples were interrogated in wells A03 to P22. The high-throughput data software Mscreen was utilized for the primary hit, validation selection, and analysis of concentration-response curve results (1). Following compound addition, cells were cultured for 48 h at either 5% $CO_2$ at 37 °C for HCT15 cells or atmospheric air at 37 °C for SW-48 cells. Cell viability was measured using a CellTiter-Glo luminescent kit (catalog no. G7571) from Promega as directed using a PHERAstar instrument from BMG Labtech.

## Estimating false discovery rate

The false discovery rate (FDR) is estimated with a target-decoy approach. We select nystatin as the testing molecule. The target database is generated by applying all the detected modifications to the correct core structures. Decoys are generated via a fixed number of edge-switching steps on target database molecules. The edge-switching operation selects two edges in a graph and randomly swaps their endpoints. To operate on molecules, we model a molecule as a connected multigraph and enforce that there is only a single connected component after an edge-switching operation. We generated decoys for the target database by applying 25 edge-switching steps on each compound and filtering out generated decoys that appear in the target database by InChIKey. The *P* value for all the matching in the decoy and target database is shown in Supplementary Figs. 22 and 23. Given a p-value threshold, let $N_{decoy}$ and $N_{target}$ denote the number of peptide-spectrum matches in the decoy database and the target database, respectively. The FDR can be estimated as follows:

$$FDR = \frac{N_{decoy}}{N_{target}} \qquad (2)$$

## Data availability

The MS datasets used in this study are publicly available from the GNPS infrastructure under the following accession code: MSV00083738. The

MS data for *Streptomyces cinnamoneus* NRRL B-24434 has been uploaded to MassIVE under the following accession code: MSV000094063. The MS data for *Streptomyces actiphen* has been uploaded to MassIVE under the following accession code: MSV000094894.

## Code availability

Seq2PKS is available at https://github.com/mohimanilab/Seq2PKS[51]. The Seq2PKS server is available at https://run.npanalysis.org/.

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

## Acknowledgements

D.Y., M.Z., A.A., M.G., S.L., H.S., T.K., G.O., X.L., Y.D., L.C., and H.M. were supported by National Institutes of Health New Innovator Award DP2GM137413, US Department of Energy award DE-SC0021340, and National Science Foundation award DBI-2117640. The work of B.B. was supported by the National Institute of General Medicine Sciences of the National Institutes of Health award R43GM150301. R.D.K. and X.W. were supported by NIGMS (R35GM146934), and D.H.S. by R35 GM118101. The authors would like to acknowledge the support from the University of Michigan Biosciences Initiative (A.T. and D.H.S.). The authors thank Prof. D.H. Sherman (University of Michigan–Ann Arbor) for developing the Sherman Microbial Repository and extract library. This natural product extract library is currently curated and maintained by NPDC (University of Michigan–Ann Arbor) led by Prof. Ashootosh Tripathi and the library can be accessed for screening purposes by emailing at natprodcore@umich.edu. The authors want to thank Ms. Pam Schultz and Ms. Amudha Ramchandran from NPDC for helping with fermentation and DNA extraction. The authors would like to thank Dr. Aaron Robida from Center for Chemical Genomics (University of Michigan-Ann Arbor) for helping with cell based bioactivity assessment.

## Author contributions

D.Y., M.Z., A.A., M.G., S.L., H.S., T.K., G.O., X.L., Y.D., L.C., B.B., and H.M. implemented the Seq2PKS algorithm and performed the analysis. X.W. and R.D.K. performed the experimental validation of the identified biosynthetic gene clusters for monazomycin. Y.Z. and A.T. performed the experimental validation of the identified molecule 2-aminobenzamide-lactophenol. D.Y., A.A., and M.G. designed and implemented the Seq2PKS server. D.H.S., P.J.S., R.D.K., J.A.C., A.T., B.B., and H.M. directed the work. D.Y., R.D.K., A.T., B.B., and H.M. wrote the manuscript.

## Competing interests

H.M. and B.B. are co-founders and have equity interests from Chemia Biosciences Inc. The remaining authors declare no competing interests.
