## [Peer Review File · Nature Communications]

Discovering Type I Cis-AT polyketides through computational mass spectrometry and genome mining with Seq2PKSREVIEWER COMMENTS

Reviewer #1 (Remarks to the Author):

In their manuscript, Yan et al. introduce their tool Seq2PKS, which allows automated mapping of genome mining data of modular type I polyketide synthase BGCs to mass spectrometry data and thus helps in identifying the products of the PKS BGCs.

For predicting the PKS product, Seq2PKS first identifies the PKS BGCs with antiSMASH. In a second step, the specificity of the AT domains is predicted using a decision tree algorithm. PKS docking motifs are analyzed to infer the order of the PKS assembly line. Based on these data and the presence/absence of PKS reductive domain, structures are predicted. These furthermore are expanded by applying potential tailoring/modification steps, which are inferred from tailoring enzymes also encoded in the BGC. With that approach, hundreds to thousands of potential structures are generated per gene cluster, which then can be mapped to MS data using GNPS and Dereplicator+.

The authors validated their tool by matching spectra and BGC of various known compounds and identified the BGC of two PKS, monazomycin and oasomycin, which weren't described before.

This is a highly interesting approach that will benefit the discovery of novel polyketides and supports identifying BGCs of known compounds. Especially the optimized algorithm to predict AT specificities and the inclusion of common tailoring reactions is an important step in improving the predictive capabilities.

However, there are some issues which need to be addressed.

Main points:

- Unfortunately, the authors oversell the novelty of their approach in various aspects:

- Predicting PKS core structures based on modification domains of the PKS is also

implemented in many other tools (Just one of many examples

(http://202.54.226.228/~pkssdb/sbspks_updated/master.html), SBSPKS, which is based on

SEARCHPKS from 2003 (Yadav, G., et al. (2003). SEARCHPKS: A program for detection and analysis of polyketide synthase domains. *Nucleic Acids Res.* 31, 3654-3658.

It is of course highly acknowledged that the authors add a larger set of tailoring modifications in their approach, but these original approaches to predict PKS products should be acknowledged.

- The idea to calculate a large set of potential structures from PKS BGCs and map these to MS data was already implemented by the Magarvey group in their “Genomes to Natural Products” platform (Johnston, C.W et al. (2015). Nat. Commun. 6, 8421.

10.1038/ncomms9421); further the prediction of structures was refined in their PRISM tool (Skinnider, M.A., et al. (2020). Nat. Commun. 11, 6058. 10.1038/s41467-020-19986-1).

- Defining the order of the assembly line by correlating docking domains using a very similar approach was already introduced by Anand et al. (Anand, S., et al. (2010). SBSPKS: structure-based sequence analysis of polyketide synthases. Nucleic Acids Res. 38, W487-496. DOI: 10.1093/nar/gkq340. in SBSPKS and is also implemented in commonly used tools like antiSMASH.

- The possibility to integrate genomic and MS data makes Seq2PKS software a very interesting and valuable tool for the PKS community. However, the documentation and installation instructions need to be significantly improved – at the moment deep knowledge of the pipeline and the underlying tools are required (e.g. the tool transeq mentioned in the requirements is part of EMBOSS suite (which is not mentioned, so to install transeq one needs to install EMBOSS instead of the named tool), same for sk-learn which is part of scikit-learn.

Futhermore, it seems that the software only works with specific, older versions of python==3.8; numpy==1.19.5 and scikit-learn==0.22.1. Especially for scikit-learn this is crucial, as serialized models generated with this version may not work / provide wrong results when used in other versions. Even with some experience in getting bioinformatics tools to work, I finally was unable to get the full pipeline to run without errors. A webserver (which apparently is planned according to the readme file) would be very helpful and avoid these complex installation issues for the majority of users.)

The webserver mentioned in the manuscript (page 21) was not working/accessible at the time of review.

- I feel it a bit misleading to compare performance “correct structure prediction” of a tool

that generates hundreds to thousands of possible structures and then filters with metabolomics data (which in most cases in genome mining studies is not readily available and only works for BGCs that are expressed) with tools that do these predictions solely on sequence data.

Specific comments:

- Figure 1: I find this figure confusing.
- The genes are oriented as in the BGC (i.e. + and – strand); the PKS modules however are always displayed in N- to C-terminal order
- Please check the AT domain specificity assignments; to my best knowledge, M1 should be mmal; M3 should be mal, M6 should be mmal,... I am not aware that the avermectin displayed in the example uses methoxymalonyl-CoA – this is also not reflected in the sub-structures shown.
- Figure 3: In my opinion, this figure does not provide much information that go beyond text-book knowledge of type I PKS.
- If the authors need to save figure space, Fig 4 could be moved to SI, as the key concept behind re-ordering the subunits is also displayed in Fig. 5.
- Page 3, end of 1st paragraph: “However, existing approaches fail to address these intrinsic modifications.” This is wrong. For example, antiSMASH has been using the exact same rules the authors describe on page 4, second last paragraph, since version 2 (Blin, K., et al., (2013) *Nucleic Acids Res.* 41, W204-W212. DOI 10.1093/nar/gkt449)
- Page 10: Some of the mention tools were never designed to predict final compound structures, this is like saying Seq2PKS fails to predict NRPS structures.
- Page 13, 1st paragraph: this sounds like only loading-module-free PKses are predicted correctly, and otherwise the software relies on Dereplicator+ to fill in the gaps? Is this

correctly interpreted?

- Page 17, 2nd paragraph: Seq2PKS isn't the first tool to link metabolomic and genomic data. Other tools (that use a slightly different approach by using GCFs instead of predicted structures, but still connect genomics and metabolomics data) are, for example, Hjorleifsson et al, (2021). Ranking microbial metabolomic and genomic links in the NPLinker framework using complementary scoring functions. PLoS Comput. Biol. 17, e1008920. DOI: 10.1371/journal.pcbi.1008920 or Leao, T.F. et al. (2022). NPOmix: A machine learning classifier to connect mass spectrometry fragmentation data to biosynthetic gene clusters. PNAS Nexus 1, pgac257. 10.1093/pnasnexus/pgac257.
- Page 17, 2nd paragraph: In strict sense, three, not five polyketides were identified, the other two relied on Dereplicator+ to bridge the gap
- Page 17, 3rd paragraph: To state that antiSMASH and PRISM struggle with correctly ordering PKSs is not generally correct in my experience. Using the authors' example of nystatin (GenBank entry AF263912.1, it's unclear to me if that was the version the authors used), the current antiSMASH version predicts the rearrangement just fine.
- Page 17, 4th paragraph: The substrate prediction methodology is new, the rule-based approach to account for the reductive domains in a module isn't.
- Page 17, 5th paragraph: PKS linker matching approach has been out there for years, it is potentially incrementally improved by giving a slight bonus to colinear sections, but given that the authors keep wrongly claiming that other tools can't deal with non-colinear PKSs at all I would like to see some examples where this actually makes a difference (see also general comment above).
- Page 18, substrate prediction methodology: BLAST for aligning to a single reference seems an interesting choice for active site extraction. Most established tools tend to align to a set of references in a multiple sequence alignment, either using a tool like Muscle or going via profile Hidden Markov Models (pHMMs) using a tool like hmmer. Can the authors elaborate as to why they decided to deviate from the common practice?

Reviewer #2 (Remarks to the Author):

The authors have introduced an innovative tool called Seq2PKS, which utilizes gene clusters and mass spectrometry analysis to predict Type 1 polyketide compounds. This study offers valuable insights into the realm of natural product discovery. Given the significance of the research, this work merits publication pending the resolution of the following minor inquiries:

1. In Figure 2b, please clarify the representation of colors in the confusion matrix. Does the color correspond to the number of data points? However, it seems that there are decimal values present.
2. The article mentions the utilization of the extra tree algorithm for predicting the substrate specificity of AT domains. The method involves a training set comprising 624 known substrate labels and employs 5-fold cross-validation. The corresponding results for model evaluation should be provided. Additionally, in the prediction of gene order in the skeleton assembly pathway, the training set encompasses 90 annotated pathways of Cis-AT BGCs, while the testing set consists of 80 pathways obtained from a literature search, is there any overlap between these two sets? Please note clearly.
3. Regarding the "Benchmarking against Prism and antiSMASH" section, it is noted that Seq2PKS identified 8 molecules from 80 test cases at a tanimoto similarity threshold of 1. The achieved accuracy does not appear to be outstanding. Furthermore, in terms of predicting the product, it holds significance if the prediction matches the true structure exactly, with similarity holding lesser significance.
4. The authors present two potential novel BGCs (Monazomycin and Oasomycin) identified by Seq2PKS. Could you provide some insights into the feasibility for experimentally verifying the products of these two BGCs?
5. In the comparative analysis with existing methods such as Prism and antiSMASH, please explain how can get the advantages by Seq2PKS, if possible that the underlying reasons should be elucidated with data to support the Seq2PKS's enhancements in key metrics.

6. A more detailed ReadMe file should be given for user to access/read the codes.

Reviewer #3 (Remarks to the Author):

In their paper, Yan et al. present a new computational pipeline for improving the prediction of the structures of compounds derived from polyketide synthases (PKS) and their direct association with mass spectrometry (MS) data. This pipeline has the potential to enhance the automated structure prediction of natural products originating from biosynthetic gene clusters (BGCs). However, the manuscript's current readability may not cater to a broad audience, especially within the context of Nature Communications.

The manuscript requires a more substantial background in both biosynthetic logic and the computational aspects of PKS systems for a comprehensive understanding. The Results and Methods sections are concise and would benefit from further clarification regarding the rationale behind specific steps and the advantages they confer.

Additionally, the discussion section could be improved by comparing the proposed tool with existing tools like NPlinker, which also aim to link BGCs with their respective natural products. The manuscript does not provide clear information about the overlap between the training and test datasets used in the pipeline. It is essential to ensure there is no such overlap to ensure the reliability of the results.

While the identification of putative gene clusters associated with the presented molecules is promising, the lack of experimental validation leaves uncertainty about whether these identified BGCs indeed encode the suggested compounds. Given the availability of existing structure prediction programs, it is not entirely clear what unique advantages the new pipeline offers.

Furthermore, it is crucial to consider the substantial number of potential structures that can be generated from a single structural template and assess the potential for false-positive identifications in spectral data. Unfortunately, an attempt to test the server, as recommended on the GitHub page, was hindered by its unavailability.

In summary, the tool introduced in this manuscript holds promise, but the paper requires more detailed explanations and additional context to be more accessible and impactful in the scientific community.

Reviewer #1 (Remarks to the Author):

In their manuscript, Yan et al. introduce their tool Seq2PKS, which allows automated mapping of genome mining data of modular type I polyketide synthase BGCs to mass spectrometry data and thus helps in identifying the products of the PKS BGCs.

For predicting the PKS product, Seq2PKS first identifies the PKS BGCs with antiSMASH. In a second step, the specificity of the AT domains is predicted using a decision tree algorithm. PKS docking motifs are analyzed to infer the order of the PKS assembly line. Based on these data and the presence/absence of PKS reductive domain, structures are predicted. These furthermore are expanded by applying potential tailoring/modification steps, which are inferred from tailoring enzymes also encoded in the BGC. With that approach, hundreds to thousands of potential structures are generated per gene cluster, which then can be mapped to MS data using GNPS and Dereplicator+.

The authors validated their tool by matching spectra and BGC of various known compounds and identified the BGC of two PKS, monazomycin and oasomycin, which weren't described before.

C 1.1 This is a highly interesting approach that will benefit the discovery of novel polyketides and supports identifying BGCs of known compounds. Especially the optimized algorithm to predict AT specificities and the inclusion of common tailoring reactions is an important step in improving the predictive capabilities.

However, there are some issues which need to be addressed.

R1.1 Thank you for your insightful comments. We have addressed your comments as detailed below. The revisions have been highlighted in red in both the main manuscript and the Supplementary material. Specifically, we have redefined the major contributions of Seq2PKS to emphasize our optimized algorithm for AT domain specificity prediction and the comprehensive database of tailoring reactions we constructed. Additionally, we have clarified the distinct purpose of Seq2PKS in comparison to tools like SBSPKS and antiSMASH, highlighting its unique focus on predicting the mature compound rather than the core (intermediate) structure.

Major Comments

C1.2 Predicting PKS core structures based on modification domains of the PKS is also implemented in many other tools (Just one of many examples (http://202.54.226.228/~pkfdb/sbspks_updated/master.html), SBSPKS, which is based on SEARCHPKS from 2003 (Yadav, G., et al. (2003). SEARCHPKS: A program for detection and analysis of polyketide synthase domains. Nucleic Acids Res. 31, 3654-3658).

It is of course highly acknowledged that the authors add a larger set of tailoring modifications in their approach, but these original approaches to predict PKS products should be acknowledged.

R1.2 We totally agree with the reviewer, and we have added the following sentences to the introduction sections to acknowledge the contributions of these methods:

Prior methods such as SBSPKS [<https://doi.org/10.1093/nar/gkx344>] and SEARCHPKS [<https://doi.org/10.1093/nar/gkg607>] construct a database of modification domains and their corresponding tailoring modifications for predicting the core structure of polyketides.

Seq2PKS enhances structure prediction reliability by incorporating a more comprehensive tailoring modification database and uses a graph-based approach for searching reaction sites and applying modifications to the constructed core structure of mature compounds.

C 1.3 The idea to calculate a large set of potential structures from PKS BGCs and map these to MS data was already implemented by the Magarvey group in their “Genomes to Natural Products” platform (Johnston, C.W et al. (2015). Nat. Commun. 6, 8421. 10.1038/ncomms9421); further the prediction of structures was refined in their PRISM tool (Skinnider, M.A., et al. (2020). Nat. Commun. 11, 6058. 10.1038/s41467-020-19986-1).

R1.3 The main contribution of Seq2PKS is its superior accuracy in domain specificity prediction and its application of extensive tailoring modifications. As a result, it achieves a much higher Tanimoto similarity to correct structure compared to PRISM (Seq2PKS achieves Tanimoto of 1.0 for multiple known polyketides, while the highest Tanimoto achieved by PRISM is below 0.7). Additionally, in contrast to the “Genomes to Natural Products” platform developed by Magarvey’s group that utilizes exact mass spectrometry search, Seq2PKS relies on Dereplicator+ that can perform variable dereplication search, allowing it to identify a PKS even when a substrate or a modification is mispredicted. We have added the following sentences to the introduction section to acknowledge the methods from Magarvey’s group:

Over the past two decades, various methods have been developed for polyketide genome mining and chemical structure prediction. Notably, methods like PRISM [<https://doi.org/10.1038/s41467-020-19986-1>] and the Genomes to Natural Products (GNP) platform [<https://doi.org/10.1038/ncomms9421>] generate a large set of potential structures from PK BGCs, and then map them to mass spectrometry data.

Additionally, Seq2PKS uses a variable mass spectral database search to discover novel polyketides even when a mis-predicted chain extension substrate or modification exists.

C 1.4 Defining the order of the assembly line by correlating docking domains using a very similar approach was already introduced by Anand et al. (Anand, S., et al. (2010). SBSPKS: structure-based sequence analysis of polyketide synthases. Nucleic Acids Res. 38, W487-496. DOI: 10.1093/nar/gkq340. in SBSPKS and is also implemented in commonly used tools like antiSMASH.

R1.4 While SBSPKS performs very well for some polyketides in assembly line order prediction, it has several drawbacks: (i) the algorithm assumes that the structure of the reference docking domains can be applied to other docking domains, which is often not the case, as demonstrated by several studies [<https://doi.org/10.1016/j.chembiol.2013.09.015>, <https://doi.org/10.1002/anie.201902571>]; (ii) this method results in a large number of tied ranks in the prediction results; (iii) The output of the method is a raw score, making its reliability difficult to interpret. Seq2PKS overcomes these limitations by introducing a machine learning algorithm that calculates the probability of each docking domain and selects the top pathways with the highest probability. This approach offers several advantages: (i) by including 90 training samples in the database, the algorithm can accommodate different docking domain structures; (ii) the rank is calculated based on probabilities, which avoids ties and facilitates the selection of top candidates; (iii) Compared to existing algorithms, Seq2PKS rewards the co-linearity of substrate order in the molecule to module order in the BGC, significantly improving prediction accuracy. We have added the following text to detail this:

During the biosynthesis of Cis-AT polyketides, the arrangement of genes in the assembly pathway often deviates from their linear order (e.g. non-co-linear) on the polyketide synthase (PKS) BGC. The state-of-the-art SBSPKS method introduced by Khater et al. [<https://doi.org/10.1093/nar/gkx344>] utilizes reference docking domain structures and calculates the raw affinity score for pathway predictions. This approach suffers from several limitations: (i) the algorithm assumes that the structure of the reference docking domains can be applied to other docking domains, which is often not the case, as demonstrated

by several studies [<https://doi.org/10.1016/j.chembiol.2013.09.015>,<https://doi.org/10.1002/anie.20190257>]; (ii) this method results in a large number of tied ranks in the prediction results, complicating the selection of top candidates; (iii) the output of the method is a raw score, making its reliability difficult to interpret.

Compared to other methods, the Seq2PKS method used for assembly pathway prediction includes 90 training samples in the database, which allows the algorithm to accommodate different docking domain structures. Furthermore, the algorithm assigns a probability to each order, which avoids ties and provides interpretability.

C 1.5 The possibility to integrate genomic and MS data makes Seq2PKS software a very interesting and valuable tool for the PKS community. However, the documentation and installation instructions need to be significantly improved – at the moment deep knowledge of the pipeline and the underlying tools are required (e.g. the tool transeq mentioned in the requirements is part of EMBOSS suite (which is not mentioned, so to install transeq one needs to install EMBOSS instead of the named tool), same for sk-learn which is part of scikit-learn.

Furthermore, it seems that the software only works with specific, older versions of python==3.8; numpy==1.19.5 and scikit-learn==0.22.1. Especially for scikit-learn this is crucial, as serialized models generated with this version may not work / provide wrong results when used in other versions. Even with some experience in getting bioinformatics tools to work, I finally was unable to get the full pipeline to run without errors. A webserver (which apparently is planned according to the readme file) would be very helpful and avoid these complex installation issues for the majority of users.)

The webserver mentioned in the manuscript (page 21) was not working/accessible at the time of review.

R1.5 We have addressed all the shortcomings brought up by the reviewer and provided more detailed guidelines for the manual installation of Seq2PKS (see the revised code attached). In the new version, there is no need to install EMBOSS anymore. Additionally, we have attached a docker container including all the required environments for running Seq2PKS, making it possible to run Seq2PKS without the need to set up the environment manually (see docker/seq2pks.tar attached). The guideline for using the docker image is both attached and shown in the <https://github.com/mohimanilab/Seq2PKS>. For users that still want to manually set up the environment, we provide a seq2pks.yml file from conda so the users can install the corresponding packages using conda. The sample data is also provided. Users can load and run the docker by:

```
sudo docker load -i seq2pks_docker.tar
```

```
sudo docker run -it -v $(pwd):/usr/src/app/mnt --privileged --entrypoint /bin/bash seq2pks
```

Finally, the server is now accessible at <https://run.npanalysis.org/>. Guidelines for using the server are available at <https://github.com/mohimanilab/Seq2PKS> and shown below. The sample data for testing the server is attached and provided in the GitHub page.

The Seq2PKS pipeline requires paired genomic and mass spectrometry data. After getting to the landing page, proceed to the Dashboard.

First, the user will need to create the user account.

Once the user logs in, they will be directed to the Dashboard page, where they should upload genome data and paired mass spectrometry data. Currently, we do not support Seq2PKS runs without paired mass spectrometry data, due to the high costs associated with large data streaming. To upload data, click on the Data tab, click the plus icon, select the type of data, and upload the file. The contents of the data are processed for use in downstream analysis, and the raw files are not used after this processing step.

Dashboard

Account

Projects **Data** Tasks

Projects		
ID	Name	Created

Dashboard

Account

Projects **Data** Tasks

Data				
ID	Name	Kind	Created	Status

After uploading the data, it will appear under the Data tab.

To start a new PKS workflow, the user can click on the dashboard tab. To create a new project, click the plus icon.

Next, to use the uploaded data in a project, the user should click on the project they created and link the data to the project.

Upon linking the data, it will appear in the Data section under the project. Now the user can submit a new Seq2PKS task using their data

NP Analysis

Dashboard

Account

ID: 47160ee8-db8c-45eb-90a7-5e499ebf9b5b

Created: Mar 10, 2024 @ 19:55:38

Tasks

ID	Name	Method	Created	Completed	Status
90239696-8fd8-4658-a4a4-09a1823a4c4d	chalconmycin_specific.mzXML	Mass Spectrum	Mar 10, 2024 @ 19:57:08		Success
0403075c-cb53-4106-965b-c51523d93677	chalconmycin.fasta	Genome	Mar 10, 2024 @ 19:57:08		Success
622e1903-7248-458e-8da4-0258ff9b40cd	NPAtlas	Chemical Database	Mar 10, 2024 @ 19:57:08		Success

Data

ID	Name	Kind	Created	Status
90239696-8fd8-4658-a4a4-09a1823a4c4d	chalconmycin_specific.mzXML	Mass Spectrum	Mar 10, 2024 @ 19:57:08	Success
0403075c-cb53-4106-965b-c51523d93677	chalconmycin.fasta	Genome	Mar 10, 2024 @ 19:57:08	Success
622e1903-7248-458e-8da4-0258ff9b40cd	NPAtlas	Chemical Database	Mar 10, 2024 @ 19:57:08	Success

NP Analysis

Dashboard

Account

ID: 47160ee8-db8c-45eb-90a7-5e499ebf9b5b

Created: Mar 10, 2024 @ 19:55:38

Tasks

Data

New Task

Submit a Task

Method: Seq2PKS

Input data

Genome: chalconmycin.fasta

Mass Spectrum: chalconmycin_specific.mzXML

Task Parameters

Task Name: Chalcomycin_test

Num Filtered Cores: 1000

Here the user can input several different parameters, which are described below:

Number of Filtered Cores	A hard maximum on the number of PK core structures to consider in downstream analysis.
Ion Mode	Possible ion modes for the input spectra, either positive or negative.
Precursor Ion Tolerance	Absolute error tolerance when matching precursor ions to molecules.
Product Ion Tolerance	Absolute error tolerance when matching peaks to theoretical fragments.
Max Absolute Charge	The maximum absolute charge, used to pick adducts.

Minimum Score

The minimum score for a molecule-spectrum match to be considered. Hits with scores lower than this will be silently discarded.

To submit the task, scroll down and click the Submit button. After the task is completed, the user can click on the task ID in the Tasks section of the project page to view the results of the task.

NP Analysis

Dashboard

Account

ID: 13fb3a02-22e4-4b73-966e-f268cd24e99a

Method: Seq2PKS

Processing Status: Finished

Created: Mar 10, 2024 @ 19:58:00

Completed: Mar 10, 2024 @ 19:58:15

Results

Items per page: 10 | 1-1 of 1 item | 1 of 1 page

Molecule	Scan Number	Precursor m/z	Retention Time	Molecule Exact Mass	Mass Error	Adduct	Score
<chem>CC1C(OC4C(C(OC)C(O)C(C)O4)OC)/C=CC3C(O3)C(=O)C(CC(C(OC2OC(C)CC(C2O)OC)C(C(=O)CC(O1)=O)C)C)C</chem>	5485	701.370	3194.650	700.367	-0.004	[M+H] ⁺	25

The results presented above were generated using the sample Chalcomycin BGC sequence and the corresponding mass spectrometry data as inputs. Given that this was conducted for testing purposes, the mass spectrum included only one scan that matched to Chalcomycin.

C1.6 I feel it a bit misleading to compare the performance “correct structure prediction” of a tool that generates hundreds to thousands of possible structures and then filters with metabolomics data (which in most cases in genome mining studies is not readily available and only works for BGCs that are expressed) with tools that do these predictions solely on sequence data.

R1.6 We agree with the reviewer's comment. Indeed, antiSMASH predicts structures solely based on genomics data and outputs a core structure instead of a mature structure. However, PRISM also generates many structures, similar to Seq2PKS. We added the following sentence to the caption of Figure 8 to clarify the difference between Seq2PKS and antiSMASH:

Note that Seq2PKS and PRISM generate multiple structures per BGC (which are further refined using mass spectrometry data), while antiSMASH generates a single core structure (solely based on genomics data).

Additionally, we added the following sentence to the result section:

Since antiSMASH predicts only hypothetical core structures rather than mature polyketide compounds, a lower Tanimoto similarity between antiSMASH results and the actual compounds is expected. Moreover,

Seq2PKS and PRISM predict multiple structures per BGC (to be refined by mass spectrometry in downstream steps), while antiSMASH predicts only one structure.

Moreover, we added the following sentence to the discussion section:

Existing methods, such as antiSMASH [<https://doi.org/10.1093/nar/gkad344>], are designed to predict the core structures, rather than the final compounds, of various natural products. While PRISM [<https://doi.org/10.1038/s41467-020-19986-1>] incorporates post-modifications into its analysis, its focus is primarily on different cyclizations, overlooking other complex modifications. In contrast, Seq2PKS predicts multiple mature polyketide compounds by considering a range of potential modifications and refines these predictions with mass spectrometry data.

Specific comments

C1.7 Figure 1: I find this figure confusing. The genes are oriented as in the BGC (i.e. + and – strand); the PKS modules however are always displayed in N- to C-terminal order.

Please check the AT domain specificity assignments; to my best knowledge, M1 should be mmal; M3 should be mal, M6 should be mmal,... I am not aware that the avermectin displayed in the example uses methoxymalonyl-CoA – this is also not reflected in the sub-structures shown.

R1.7 We apologize for the typo, we have now fixed the errors (see Figure below):

Figure 1: The Seq2PKS framework. This process initiates with the microbial genome, where (a) polyketide domains and enzymes are identified through genome mining (M1-M13: Modules; AT: Acyltransferase; KS: Ketosynthase; DH: Dehydratase; KR: Ketoreductase). Subsequently, (b) the specificity of the Cis-AT domains is predicted, and the corresponding substrates are determined by combining these specificities with other domains in each module (mal: malonyl-CoA; mmal: methylmalonyl-CoA). Following this, (c) the order of the assembly pathway is predicted to form the initial core structure. This leads to (d) the incorporation of post-assembly modifications. Lastly, (e) the mature structures are compared against mass spectra using Dereplicator+. The arrows illustrate the fragmentation process of the target molecule, and peaks in the mass spectra corresponding to these fragments are highlighted in red.

C1.8 Figure 3: In my opinion, this figure does not provide much information that go beyond text-book knowledge of type I PKS.

R1.8 We have moved this figure to the Supplementary material and improved its caption.

Seq2PKS utilizes a rule-based technique to predict the structure of mature PK substrates (Supplementary Figure 3). This prediction relies on the specificity of the AT domain and the functionality of other domains within each biosynthetic module, which has been extensively discussed in prior work [<https://doi.org/10.1093/nar/gkg607>, <https://doi.org/10.1093/nar/gkad344>]. Seq2PKS also identifies inactive domains and takes them into account for accurate prediction of the recruited substrates.

C1.9 If the authors need to save figure space, Fig 4 could be moved to SI, as the key concept behind re-ordering the subunits is also displayed in Fig. 5.

R1.9 We have moved Fig 4 to Supplementary materials.

C1.10 Page 3, end of 1st paragraph: “However, existing approaches fail to address these intrinsic modifications.” This is wrong. For example, antiSMASH has been using the exact same rules the authors describe on page 4, second last paragraph, since version 2 (Blin, K., et al., (2013) Nucleic Acids Res. 41, W204-W212. DOI 10.1093/nar/gkt449)

R1.10 We agree with the reviewer and have removed these sentences from the manuscript.

C1.11 Page 10: Some of the mentioned tools were never designed to predict final compound structures, this is like saying Seq2PKS fails to predict NRPS structures.

R1.11 We agree with the reviewers that antiSMASH is designed to predict the core structure instead of the final compound. We have added the below sentence in the result section to distinguish the difference between antiSMASH and Seq2PKS.

To better evaluate the power of Seq2PKS, we benchmarked the overall accuracy against the state-of-art methods PRISM [<https://doi.org/10.1038/s41467-020-19986-1>] and antiSMASH [<https://doi.org/10.1093/nar/gkad344>] (antiSMASH is not designed to predict the final compound structures) .

Since antiSMASH predicts only hypothetical core structures rather than mature polyketide compounds, a lower Tanimoto similarity between antiSMASH results and the actual compounds is expected. Moreover, Seq2PKS and PRISM predict multiple structures per BGC (to be refined by mass spectrometry in downstream steps), while antiSMASH predicts only one structure.

C1.12 Page 13, 1st paragraph: this sounds like only loading-module-free PKses are predicted correctly, and otherwise the software relies on Dereplicator+ to fill in the gaps? Is this correctly interpreted?

R1.12 Yes, the interpretation is correct. Currently, there is no accurate method for predicting the starter unit of polyketides. In contrast to other polyketide substrates, starter unit structures are very diverse. In fact, each polyketide has its unique starter units, which are shared only with homologous polyketides. This makes it infeasible to design machine learning models for accurately predicting the starter unit

structures of polyketides without overfitting to training data. To overcome this challenge, Seq2PKS relies on variable mass spectral database search methods that can identify the overall mass of the starter unit. We added the following sentence to the Discussion to clarify this point:

Currently, there is no accurate method for predicting structures of starter units. In contrast to other polyketide substrates, starter unit structures are very diverse and can even be unique to specific polyketides and their homologs. This makes it infeasible to design machine learning models for accurately predicting the starter unit of polyketides without overfitting to training data. To overcome this challenge, Seq2PKS relies on variable mass spectral database search methods that can identify the overall mass of the starter unit.

C1.13 Page 17, 2nd paragraph: Seq2PKS isn't the first tool to link metabolomic and genomic data. Other tools (that use a slightly different approach by using GCFs instead of predicted structures, but still connect genomics and metabolomics data) are, for example, Hjorleifsson et al, (2021). Ranking microbial metabolomic and genomic links in the NPLinker framework using complementary scoring functions. PLoS Comput. Biol. 17, e1008920. DOI: 10.1371/journal.pcbi.1008920 or Leao, T.F. et al. (2022). NPOmix: A machine learning classifier to connect mass spectrometry fragmentation data to biosynthetic gene clusters. PNAS Nexus 1, pgac257. 10.1093/pnasnexus/pgac257.

R1.13 We agree that Seq2PKS is not the first tool to integrate metabolomic and genomic data. As highlighted in Hjorleifsson. et al. "Ranking microbial metabolomic and genomic links in the NPLinker framework using complementary scoring functions", there are two primary types of tools used for this purpose: feature-based and correlation-based approaches. Feature-based approaches entail searching a set of MS2 spectra for structural features predicted in the product of a BGC. Correlation-based approaches, conversely, seek similarities within specific Gene Cluster Families (GCFs) in certain strains, utilizing these relationships to link GCFs with Molecular Families (MFs). Seq2PKS is a feature-based approach. We have included the following sentence in the discussion section to clarify this:

Currently, two types of tools bridge metabolomics and genomic data. Correlation-based approaches like NPLinker [<https://doi.org/10.1371/journal.pcbi.1008920>], NPOmix [<https://doi.org/10.1093/pnasnexus/pgac257>], and other metabolomics methods [<https://doi.org/10.1038/nchembio.1659>, <https://doi.org/10.1038/s41589-019-0400-9>, <https://doi.org/10.1038/s41589-023-01276-8>] focus on linking gene cluster families (GCFs) with molecular families (MFs) or spectra, based on co-occurrence of molecular features and BGCs. Conversely, feature-based approaches like GNP [<https://doi.org/10.1038/ncomms9421>], MetaMiner [<https://doi.org/10.1016/j.cels.2019.09.004>], Seq2RiPP [<https://doi.org/10.1038/s41467-023-39905-4>] and NRPminer [<https://doi.org/10.1038/s41467-021-23502-4>] strive to associate BGCs with mass spectra by predicting the hypothetical structure of BGC products, followed by in silico mass spectral database search. Seq2PKS is a feature-based method that integrates metabolomics and genomics data capabilities for the efficient and scalable identification of polyketides.

C1.14 Page 17, 2nd paragraph: In strict sense, three, not five polyketides were identified, the other two relied on Dereplicator+ to bridge the gap

R1.14 We have revised the text to clarify that two of the polyketides depended on Dereplicator+ to bridge the gap:

Searching the paired genomics and metabolomics dataset of *Streptomyces* strains with this feature, Seq2PKS correctly identified three known polyketides that lacked starter units. Two additional known

polyketides with starter units were identified using Dereplicator+ in variable search mode on the Seq2PKS outputs.

C1.15 Page 17, 3rd paragraph: To state that antiSMASH and PRISM struggle with correctly ordering PKSs is not generally correct in my experience. Using the authors' example of nystatin (GenBank entry AF263912.1, it's unclear to me if that was the version the authors used), the current antiSMASH version predicts the rearrangement just fine.

R1.15 We completely agree with the reviewer. To better compare our method with antiSMASH's results for assembly order prediction, we submitted 90 polyketide BGCs from our benchmark database (consisting of 63 co-linear and 27 non-co-linear polyketide BGCs) to the antiSMASH server to obtain the predicted order. Currently, antiSMASH considers the co-linear order by default, so it always provides the correct order for co-linear ones. Since antiSMASH cannot report orders for hybrid (NRP-PKS) BGCs, we manually removed the NRP genes from these hybrid BGCs and submitted the remaining sequences to the server. At the same time, we extracted our predicted interaction score in our five-fold cross-validation for the testing samples using the best model. In this way, we avoided sample overlap between training and testing data. Of the 27 non-co-linear samples, antiSMASH correctly predicted the assembly pathway for 20, while Seq2PKS correctly predicted 25. These results demonstrate that antiSMASH is highly accurate in predicting the correct assembly pathway, and Seq2PKS further increases accuracy by rewarding co-linearity.

We added the following paragraph to the result section to address the comparison between Seq2PKS and antiSMASH:

To compare our method with antiSMASH's predictions for assembly order, we selected all 27 non-co-linear PKSs from our benchmark dataset and predicted their assembly order using antiSMASH. Concurrently, we extracted Seq2PKS interaction scores during our five-fold cross-validation for the testing samples using the best-performing model for Seq2PKS, thus ensuring no sample overlap between training and testing data. These interaction scores were then employed to rank the assembly orders. Of the 27 non-co-linear samples, antiSMASH accurately predicted the assembly pathway for 20, while Seq2PKS correctly predicted 25 (Figure 4).

Figure 4: Comparison of pathway prediction accuracy between PNN, antiSMASH, and DDAP in (A) all 90 candidate BGCs and (B) 27 non-co-linear BGCs in the dataset. In both scenarios, PNN outperforms DDAP and antiSMASH in accuracy. Note that for antiSMASH, only the rank one order is reported.

We added the following paragraph to the discussion section:

Moreover, Seq2PKS introduces an innovative approach for predicting the assembly pathway. By considering the gene order in the genome, Seq2PKS accounts for the similarities between substrate order in polyketides and gene order in their BGCs, enabling more precise predictions of the substrate order. Also, by calculating the probabilities instead of predictions, Seq2PKS can evaluate multiple candidates from the output result.

C1.16 Page 17, 4th paragraph: The substrate prediction methodology is new, the rule-based approach to account for the reductive domains in a module isn't.

R1.16 We agree with the reviewer and have removed these sentences from the manuscript.

C1.17 Page 17, 5th paragraph: PKS linker matching approach has been out there for years, it is potentially incrementally improved by giving a slight bonus to co-linear sections, but given that the authors keep wrongly claiming that other tools can't deal with non-co-linear PKSs at all I would like to see some examples where this actually makes a difference (see also general comment above).

R1.17 We totally agree with the reviewers that antiSMASH has high accuracy in order prediction. We have performed extensive analysis in R1.15.

C1.18 Page 18, substrate prediction methodology: BLAST for aligning to a single reference seems an interesting choice for active site extraction. Most established tools tend to align to a set of references in a multiple sequence alignment, either using a tool like Muscle or going via profile Hidden Markov Models (pHMMs) using a tool like hmmer. Can the authors elaborate as to why they decided to deviate from the common practice?

R1.18 We apologize for the typo. The mention of BLAST alignment was actually a typo in our manuscript. We initially used BLAST in the earlier versions of Seq2PKS. However, to improve accuracy, we later switched to MUSCLE for active site extraction, and all our results are based on MUSCLE. We have corrected this:

For each AT-domain in our training data, we extracted 24 active site residues reported by Yadav et al. [[https://doi.org/10.1016/S0022-2836\(03\)00232-8](https://doi.org/10.1016/S0022-2836(03)00232-8)] by aligning the AT-domain to the reference sequence using MUSCLE [<https://doi.org/10.1093/nar/gkh340>] sequence alignment.

Reviewer #2 (Remarks to the Author):

C2.1 The authors have introduced an innovative tool called Seq2PKS, which utilizes gene clusters and mass spectrometry analysis to predict Type 1 polyketide compounds. This study offers valuable insights into the realm of natural product discovery. Given the significance of the research, this work merits publication pending the resolution of the following minor inquiries

R2.1 Thank you for the suggestions on our paper – we have provided a point-by-point response to each of your comments below. The changes are indicated in red in the revised manuscript and the Supplementary material.

Major Comments

C2.2 In Figure 2b, please clarify the representation of colors in the confusion matrix. Does the color correspond to the number of data points? However, it seems that there are decimal values present.

R2.2 Yes, the colors correspond to the number of data points. The presence of decimal values in the previous version of the graph was due to the values being obtained by randomly shuffling the data five times, with each shuffle followed by a five-fold cross-validation to obtain the predicted label for each sample. The results were then averaged across the five random shuffles to mitigate bias during the data-splitting process. We have included the following sentence in the manuscript to explain this procedure and updated the figure after rounding the decimal values to the nearest integer.

Figure 2 (b): Confusion matrix for extra-tree prediction. The results are averaged across five different shuffles (used in five-fold cross-validation) and rounded to the nearest integer.

C2.3 The article mentions the utilization of the extra tree algorithm for predicting the substrate specificity of AT domains. The method involves a training set comprising 624 known substrate labels and employs five-fold cross-validation. The corresponding results for model evaluation should be provided. Additionally, in the prediction of gene order in the skeleton assembly pathway, the training set encompasses 90 annotated pathways of Cis-AT BGCs, while the testing set consists of 80 pathways obtained from a literature search. Is there any overlap between these two sets? Please note clearly.

R2.3 The model evaluation metrics for assessing the substrate specificity prediction of AT domains in five-fold cross-validation have been added to supplementary material (Supplementary Figure 1 below).

Supplementary Figure 1: Evaluation metrics for the extra-tree algorithm in predicting AT domain specificity during five-fold cross-validation.

We acknowledge the importance of avoiding overlap between the training and testing sets during the model evaluation process. Therefore, the accuracy figures presented in the manuscript (Figure 4) for assembly pathway prediction are based on the 90 annotated pathways, utilizing five-fold cross-validation. The ROC-AUC curve for this cross-validation result is presented in Supplementary Figure 5. Among the 80 pathways obtained from the literature review, 61 BGCs overlap with the 90 pathways above. To avoid overlap, we have revised the manuscript to only test the model on the 19 non-overlapping BGCs and report the accuracy. One thing worth noting is that for these 19 BGCs, the entire genome sequence is used as the starting point, necessitating Seq2PKS to first identify the genes and the docking domain sequence before predicting the assembly order. This adds to the complexity of the task. The objective of this test was to mainly evaluate how Seq2PKS performs with genome sequence as the starting point. Seq2PKS correctly identified 14 out of the 19 BGCs. Among these 14, Seq2PKS correctly predicted the assembly order for 12 BGCs. To elucidate these points, we have included the following sentence in our manuscript to address the overlap in samples between different tests:

Figure 4 illustrates the five-fold cross-validation accuracy of the PNN method compared with DDAP, which employs a support vector machine algorithm [<https://doi.org/10.1093/bioinformatics/btz677>]. As a

benchmark dataset, we used docking domain sequences from 90 BGCs reported in the DDAP paper. The ROC-AUC curve for the PNN method is shown in Supplementary Figure 5.

We also tested the efficacy of PNN on 19 reported BGCs that are non-overlapping with our training data. Starting with the genome sequences containing these BGCs, Seq2PKS first identifies the genes and the docking domain sequences using HMM search before predicting the assembly order. For 14 out of 19 BGCs, Seq2PKS correctly identified the pathway genes. Among these 14, Seq2PKS correctly predicted the assembly order for 12 BGCs.

Figure 4: Comparison of pathway prediction accuracy between PNN, antiSMASH, and DDAP in (A) all 90 candidate BGCs and (B) 27 non-co-linear BGCs in the dataset. In both scenarios, PNN outperforms DDAP and antiSMASH in accuracy. Note that for antiSMASH, only the rank one order is reported.

Supplementary Figure 5: ROC-AUC curve for PNN method during five-fold cross-validation process.

C2.4 Regarding the "Benchmarking against Prism and antiSMASH" section, it is noted that Seq2PKS identified 8 molecules from 80 test cases at a tanimoto similarity threshold of 1. The achieved accuracy does not appear to be outstanding. Furthermore, in terms of predicting the product, it holds significance if the prediction matches the true structure exactly, with similarity holding lesser significance.

R2.4 The biosynthetic pathway for polyketides is long and complex, which makes predicting mature compounds challenging. Existing methods like antiSMASH and PRISM primarily focus on predicting the core structure or cyclized core structure of polyketides. However, these predictions still require extensive tailoring modifications to produce the final mature compounds. Therefore, achieving a Tanimoto similarity threshold of 1 for 8 out of 80 molecules is significant, especially considering that the highest Tanimoto similarity achieved by PRISM and antiSMASH is 0.65 and 0.30, respectively. If we consider all 116 polyketides (including those not reported in the Prism paper), Seq2PKS correctly recovers 16 molecules (Table 1). Additionally, our results show that even in cases when Seq2PKS does not make a perfect prediction, we can correct for the errors made during genome mining using Dereplicator+ in variable mode to predict modification mass and site of the error. For example, in the case of polyketides aureothin and vicenistatin, Seq2PKS prediction was 58% and 72% similar to correct structures, and by variable mass spectral database search, we were able to match their mass spectra to the correct structure. We added the following sentence to clarify this in the discussion section:

By constructing an extensive tailoring modification database and applying it to the cyclized core structure, Seq2PKS achieves the correct mature structure for 16 molecules. In the case of polyketides with uncommon starter units, Seq2PKS successfully recovers the compound structures for seven molecules, with the exception of the starter units.

Existing methods, such as antiSMASH [<https://doi.org/10.1093/nar/gkad344>], are designed to predict the core structures, rather than the final compounds, of various natural products. While PRISM [<https://doi.org/10.1038/s41467-020-19986-1>] incorporates post-modifications into its analysis, its focus is primarily on different cyclizations, overlooking other complex modifications. In contrast, Seq2PKS predicts multiple mature polyketide compounds by considering a range of potential modifications and refines these predictions with mass spectrometry data.

Overall, Seq2PKS correctly predicts eight out of 80 polyketides in the benchmark dataset (16 if counting polyketides without PRISM result), while PRISM and antiSMASH do not predict any of the polyketides correctly. The highest Tanimoto similarity that PRISM and antiSMASH achieve are 0.65 and 0.30.

C2.5 The authors present two potential novel BGCs (Monazomycin and Oasomycin) identified by Seq2PKS. Could you provide some insights into the feasibility for experimentally verifying the products of these two BGCs?

R2.5 We thank the reviewer for bringing this up. We used two methods to validate the products. Due to the unavailability of the oasomycin A standard, our analysis primarily focused on the validation of the monazomycin BGC. First, we cultured *Streptomyces cinnamoneus* NRRL B-24434 in ISP2 medium. This was followed by extraction of the fermentation broth with 1-butanol, rotary evaporation of the 1-butanol phase, and resuspension of the extract in methanol. We then performed LC-MS (Liquid Chromatography-Mass Spectrometry) analysis on the 1-butanol extract. The LC-MS results, compared with an authentic monazomycin standard from Santa Cruz Biotechnology, are depicted in the figure below (Supplementary Figure 11,12). The retention time and mass spectrum observed from our samples matched that of the authentic standard, confirming that the strain indeed produces monazomycin as expected. We also annotated the BGC of monazomycin to ensure that the machinery for the production of the polyketides is present, as shown below (Supplementary Tables 1). We have added the following sentence to the results section:

Experimental validation of monazomycin BGC. To further corroborate the novel BGC identified for monazomycin, we extensively annotated the BGC (Supplementary Table 1). The annotation results confirmed the necessary biosynthetic machinery to produce monazomycin.

We cultivated *Streptomyces cinnamoneus* NRRL B-24434 and collected LC-MS/MS data on the butanolic extracts of its growth medium. Comparison with the authentic standard in both retention time (Supplementary Figure 11) and tandem mass spectrometry data (Supplementary Figure 12) was consistent with monazomycin production by *Streptomyces cinnamoneus* NRRL B-24434.

Supplementary Table 1: Gene annotation for identified monazomycin BGC in *Streptomyces cinnamoneus* NRRL B-24434.

	Location	Nucleotide	Amino Acid	Hypothesized Function
Contig 1	2 - 1,417	1,416	472	LuxR family transcriptional regulator
Contig 2	1,573 - 2,337	765	254	alpha/beta fold hydrolase (thioesterase)
Contig 3	3,173 - 5,959	2,787	928	LuxR family transcriptional regulator
Contig 4	6,241 - 7,899	1,659	552	NAD(P)/FAD-dependent oxidoreductase
Contig 5	8,450 - 9,154	705	234	thioesterase
Contig 6	9,280 - 10,692	1,413	470	class I adenylate-forming enzyme family
Contig 7	10,751 - 11,701	951	316	ACP S-malonyltransferase
Contig 8	11,733 - 13,025	1,293	430	histidine kinase
Contig 9	13,359 - 14,900	1,542	513	glycosyltransferase
Contig 10	14,973 - 16,214	1,242	413	cytochrome P450
Contig 11	16,343 - 19,726	3,384	1,127	SDR family NAD(P)-dependent oxidoreductase
Contig 12	20,001 - 29,660	9,660	3,219	type I polyketide synthase
Contig 13	29,748 - 45,353	15,606	5,201	type I polyketide synthase
Contig 14	45,420 - 56,489	11,070	3,689	type I polyketide synthase
Contig 15	56,536 - 71,967	15,432	5,143	type I polyketide synthase
Contig 16	72,473 - 86,383	13,911	4,636	type I polyketide synthase
Contig 17	86,428 - 107,196	20,769	6,922	type I polyketide synthase
Contig 18	107,219 - 125,716	18,498	6,165	type I polyketide synthase
Contig 19	126,193 - 138,252	12,060	4,019	type I polyketide synthase
Contig 20	138,360 - 138,524	165	55	unknown

Supplementary Figure 11: LCMS-based comparison of monazomycin analyte from our cultured strain *Streptomyces cinnamoneus* NRRL B-24434 and an authentic standard provided by Santa Cruz Biotechnology.

Supplementary Figure 12: Tandem mass spectra for monazomycin from our cultured strain *Streptomyces cinnamoneus* NRRL B-24434 (top) and authentic standard (bottom) provided by Santa Cruz Biotechnology.

Additionally, we have incorporated the following sentence into the methods section:

Validation of monazomycin BGC. We cultivated liquid cultures of *Streptomyces cinnamoneus* NRRL B-24434 in ISP2 medium for 7 days at 28°C. The shaking speed is 220 rpm with a flask size of 1L and medium content of 100mL. This was followed by extraction of the broth with 1-butanol, rotary evaporation of the 1-butanol extract, resuspension of the extract in 80% methanol, followed by LC-MS/MS analysis. LC-MS data was collected on a Thermo QExactive Orbitrap connected to a Vanquish LC system. LC settings were as follows: injection volume 5 µl; Phenomenex Kinetex 2.6 µm C18 reverse phase 100 Å 150 mm x 3 mm LC column; LC gradient, solvent A, 0.1% formic acid; solvent B, acetonitrile (0.1% formic acid); 0 min, 10% B; 5 min, 60% B; 5.1 min, 95% B; 6 min, 95% B; 6.1 min, 10% B; 9.9 min, 10% B; 0.5 ml/min. MS settings were as follows: positive ion mode; full MS, resolution 70,000; mass range 400-1,200 m/z; dd-MS2 (data-dependent MS/MS), resolution 17,500; AGC target 1×10^5 , loop count 5, isolation width 1.0 m/z, collision energy 25 eV, dynamic exclusion 0.5 s.

In addition, we identified a novel actiphenol variant 2-aminobenzamide-actiphenol using Seq2PKS. The finding was validated with extensive experiments. This finding further proves the power of Seq2PKS in discovering novel compounds. We have added below sentence in the result section to address this novel finding.

Identification of novel actiphenol variant 2-aminobenzamide-actiphenol. We applied Seq2PKS to the *Streptomyces actiphen* genome available through the University of Michigan Natural Product Discovery Core and identified a long PK system with 55% similarity to the previously proposed actiphenol BGC. Seq2PKS generated one core structure, resulting in 258 mature compounds for this PK BGC. By searching the constructed molecules in a spectral dataset using Dereplicator+, we identified two PK-spectral matches with scores of 21 and 28. We extensively annotated the BGC for these molecules (Supplementary Table 2) and compared it with the reported BGC for actiphenol from MiBIG database (Supplementary Figure 13). The former molecule is a previously reported variant of actiphenol called Nong-Kang 101-G (Figure 9) [http://sioc-journal.cn/Jwk_hxxb/EN/abstract/article_342462.shtml], while the

latter is a novel actiphenol congener in which the cyclohexanone unit is substituted by a phenol moiety (Figure 10) [<https://doi.org/10.1038/nchembio.304>].

The new actiphenol molecule was isolated as a yellow solid with a molecular formula of $C_{22}H_{22}N_3O_5$, derived from an HRESIMS ion peak of $C_{22}H_{22}N_3O_5$ [M-H]⁻ (*m/z*: found 408.1455, calcd 408.1559; NMR data for the new molecule is shown in Supplementary Figures 14, 15, 16, 17, 18, 19 and Supplementary Table 3). Careful analyses of the 1D and 2D NMR spectra, confirmed the actiphenol moiety, however, the C-4 methyl group is absent. Instead, we observed a new methylene ($\delta_H = 4.30$) at C-15 ($\delta_C = 45.9$) that correlates with C-3 ($\delta_C = 136.9$), C-5 ($\delta_C = 127.6$), and an aromatic carbon C-16 (chemical shift of ($\delta_C = 149.9$)) atypical of the actiphenol spin systems. Further examination revealed the presence of an additional aminobenzamide moiety attached to the actiphenol through an N-C bond. The HMBC correlations between the triplet H-18 ($\delta_H = 7.23$)/C-16 ($\delta_C = 149.9$) and the doublet H-20 ($\delta_H = 7.61$)/C-16 ($\delta_C = 149.9$) and C-22 ($\delta_C = 171.9$) confirmed the relative position of the secondary amine ($\delta_H = 7.23$) and the primary amide on this aromatic ring. Based on its structure, we named this molecule 2-aminobenzamide-actiphenol (Figure 11).

We tested the molecules against two cancer cell lines, SW48 (CCL231, colon cancer) and HCT15 (CCL-225 colon cancer) based on the initial strong cytotoxic activity observed (Supplementary Figure 20) from the crude extracts generated by the producing strain (*Streptomyces actiphen*) during the initial isolation. However, the isolated novel molecule 2-aminobenzamide-actiphenol did not show appreciable activity upon isolation and characterization (Supplementary Figure 21)

Figure 9: Identification of Nong-Kang 101-G by Seq2PKS. (a) Domains and modules in the biosynthetic genes are identified. (b) Substrate specificity and mature substrate for each module are predicted. (c) Assembly order is predicted, and the core structure is constructed by connecting the predicted mature substrates for each module. (d) Cyclization and hydroxylation modifications are applied by two enzymes from the BGCs (shown in pink and orange, respectively). (e) Fragments from the hypothetical molecule (shown in black) that match a peak in mass spectra are highlighted (shown in red). Fragments are generated by one or two rounds of fragmentation of the hypothetical molecule. (f) The annotated spectrum of the hypothetical molecule is shown in red.

Figure 10: Identification of 2-aminobenzamide-actiphenol by Seq2PKS. (a) Domains and modules in the biosynthetic genes are identified. (b) Substrate specificity and mature substrate for each module are predicted. (c) Assembly order is predicted, and the core structure is constructed by connecting the predicted mature substrates for each module. (d) Cyclization and dehydroxylation modifications are applied by two enzymes from the BGCs (shown in pink and orange, respectively). While Seq2PKS cannot predict the presence of 2-aminobenzamide, it correctly predicts its molecular formula from the difference between the precursor mass spectrum and the total mass of the hypothetical molecule. (e) Fragments from the hypothetical molecule (shown in black) that match a peak in mass spectra are highlighted (shown in red). Fragments are generated by one or two rounds of fragmentation of the hypothetical molecule. (f) The annotated spectrum of the hypothetical molecule is shown in red.

Chemical Formula: C₂₂H₂₃N₃O₅

Exact Mass: 409.16

Figure 11: ¹H NMR Spectral Analysis showcasing the chemical shifts and multiplicity patterns for the identification of 2-aminobenzamide-actiphenol.

Supplementary Table 2: Gene annotation for 2-aminobenzamide-actiphenol BGC.

	Location	Nucleotide	Amino Acid	Hypothesized Function
Contig 1	2 - 901	900	299	Polar amino acid ABC transporter
Contig 2	905 - 1,657	753	250	ABC transporter
Contig 3	1,753 - 2,538	786	261	Unknown
Contig 4	2,608 - 3,297	690	229	GntR family transcriptional regulator
Contig 5	3,343 - 4,149	807	268	Short-chain dehydrogenase/reductase SDR
Contig 6	4,370 - 5,017	648	215	Unknown
Contig 7	5,017 - 5,667	651	216	Unknown
Contig 8	5,664 - 6,539	876	291	ABC transporter
Contig 9	6,536 - 7,639	1,104	367	Transport system permease protein
Contig 10	7,620 - 8,774	1,155	384	Iron compound ABC transporter
Contig 11	9,012 - 9,158	147	48	Unknown
Contig 12	9,460 - 10,227	768	255	Unknown
Contig 13	10,475 - 11,200	726	241	Unknown
Contig 14	11,658 - 12,803	1,146	381	Unknown
Contig 15	12,858 - 13,307	450	149	Unknown
Contig 16	14,117 - 14,467	351	116	Unknown
Contig 17	14,738 - 15,448	711	236	Unknown
Contig 18	16,234 - 16,437	204	67	Unknown
Contig 19	16,458 - 17,060	603	200	Unknown
Contig 20	17,162 - 17,626	465	154	Unknown
Contig 21	17,662 - 18,789	1,128	375	Serine/threonine protein kinase
Contig 22	18,966 - 19,784	819	272	Unknown
Contig 23	20,001 - 40,511	20,511	6,836	Type I polyketide synthase
Contig 24	40,604 - 42,643	2,040	679	Asparagine synthase
Contig 25	42,660 - 42,917	258	85	Putative acyl carrier protein
Contig 26	42,921 - 43,796	876	291	Type I polyketide synthase
Contig 27	44,081 - 47,410	3,330	1,109	Transcriptional regulator
Contig 28	47,582 - 50,980	3,399	1,132	NRP synthase
Contig 29	51,124 - 52,323	1,200	399	Cytochrome P450
Contig 30	52,477 - 53,262	786	261	Short-chain dehydrogenase/reductase SDR
Contig 31	53,900 - 54,988	1,089	362	Flavin oxidoreductase
Contig 32	55,091 - 56,020	930	309	LysR family transcriptional regulator

Supplementary Table 3: NMR result for 2-aminobenzamide-actiphenol.

Position	δ_c (type)	δ_H , multiplets (J in Hz)	COSY	HMBC	ROESY
1	159.3 (C)				
1-OH		12.33, s		1, 2, 6	
2	127.0 (C)				
3	136.9 (CH)	7.45, s		1, 5, 14, 15	14, 15a, 15b
4	130.0 (C)				
5	127.6 (CH)	7.82, s		1, 3, 7, 15	8a, 8b, 15a, 15b
6	118.7 (C)				
7	205.6 (C)				
8a	42.9 (CH ₂)	3.18, d (6.4)	9	7, 9, 10, 13	5, 10a, 10b, 13a, 13b
8b	42.9 (CH ₂)	3.18, d (6.4)	9	7, 9, 10, 13	5, 10a, 10b, 13a, 13b
9	26.6 (CH)	2.62, m	8a, 8b		
10a	37.5 (CH ₂)	2.39, dd (16.5, 10.7)	10b	11, 12	5
10b	37.5 (CH ₂)	2.58, dd (16.5, 10.7)	10a	11, 12	5
11	173.4 (C)				
11-NH		10.76, s		10, 13	
12	173.4 (C)				
13a	37.5 (CH ₂)	2.39, d (16.5, 10.7)	13b	11, 12	5
13b	37.5 (CH ₂)	2.58, d (16.5, 10.7)	13a	11, 12	5
14	15.8 (CH ₃)	2.17, s		1, 2, 3	3
15a	45.9 (CH ₂)	4.30, s		3, 4, 5, 16	3, 5, 17
15b	45.9 (CH ₂)	4.30, s		3, 4, 5, 16	3, 5, 17
15-NH		8.47, br s			
16	149.9 (C)				
17	112.1 (CH)	6.69, d (8.4)	18	19, 21	15a, 15b, 18
18	132.8 (CH)	7.23, t (7.9)	17, 19	16, 20	17, 19
19	114.9 (CH)	6.54, t (7.5)	18, 20	17, 21	18, 20
20	129.5 (CH)	7.61, d (7.8)	19	16, 18, 22	18, 19
21	114.9 (C)				
22	171.9 (C)				
22-NHa		7.17, br s			
22-NHb		7.86, br s			

Supplementary Figure 13: Comparison of Biosynthetic Gene Clusters (BGCs) between the 2-aminobenzamide-actiphenol and actiphenol from the MIBiG database (BGC0000175).

Supplementary Figure 14: ¹H spectrum of 2-aminobenzamide-actiphenol.

Supplementary Figure 15: ^{13}C spectrum of 2-aminobenzamide-actiphenol.

Supplementary Figure 16: COSY spectrum of 2-aminobenzamide-actiphenol.

Supplementary Figure 17: HSQC spectrum of 2-aminobenzamide-actiphenol.

Supplementary Figure 18: ^1H - ^{13}C spectrum of 2-aminobenzamide-actiphenol.

Supplementary Figure 19: ROESY spectrum of 2-aminobenzamide-actiphenol.

44321_A2I-80L-Crude

Supplementary Figure 20: Activity of crude extracts generated from 2-aminobenzamide-actiphenol producing strain *Streptomyces actiphen* against SW48 (CCL231, Colon Cancer) and HCT15 (CCL-225 Colon Cancer) cell line.

2-aminobenzamide-actiphenol (20)

Supplementary Figure 21: Activity of the isolated novel molecule 2-aminobenzamide-actiphenol against SW48 (CCL231, Colon Cancer) and HCT15 (CCL-225 Colon Cancer) cell line.

We added below part to the method section to discuss the experimental protocol:

Genome Extraction, Sequencing, Assembly, and Annotation. The genomic DNA for 44321-A2 was extracted using BIOSEARCH Technologies MasterPure Complete DNA and RNA purification kit following the manufacturer's protocol with some modifications, including additional lysis steps using EDTA (50mM) and lysozyme (10mg/mL) and one more step of heat treatment at 95 °C before RNase A treatment. The genomic DNA was sequenced at the University of Minnesota, Genomics Center, Minneapolis, MN. The sample library was prepared using PacBio Sequel II HiFi -- the SMRT Cell 8M typically generates approximately 4--5 million raw reads with flexible sequencing run times of up to 30 hours, yielding a 1.04 GB fastq file. Demultiplexing and quality control were done using PATRIC service which obtained a total of 90,490 read pairs. The 44321-A2 was assembled using Canu version 1.7.1 and 2 rounds of polishing

done as iteration using Racon version 2.4.13 as a part of the comprehensive genome analysis service at PATRIC. QUAST version 5.0.2, minimap2 (2.17-r974-dirty), samtools Version 1.11, and Bandage 0.8.1 with default parameters were used for assembly quality assessment and visualization. The resulting assembled genome has an estimated length of 8,799,402 bp and an average GC content of 72.18%. Two contigs generated for this genome have 7,647 protein coding sequences (CDS), 68 transfer RNA (tRNA) genes, and 18 ribosomal RNA (rRNA) genes. The annotation included 2,918 hypothetical proteins and 4,729 proteins with functional assignments. The proteins with functional assignments included 1,270 proteins with Enzyme Commission (EC) numbers, 1,111 with Gene Ontology (GO) assignments, and 1,007 proteins that were mapped to KEGG pathways. Then the assembled contig FASTA file is used to extract the 16S fragments with the help of ContEst16S tool at Ezbiocloud. Whole-genome similarity metrics, including average nucleotide identity (ANI) and DNA–DNA hybridization (DDH), were obtained to estimate genetic relatedness and define phylogeny. FastANI showed 86% genome-relatedness of our microbial strain genome to *Streptomyces atratus* ASM333086v1 and *Streptomyces gelaticus* ASM1464953v1. A whole-genome-sequence-based phylogenetic tree was built using the TYGS analysis method further supporting the genetic closeness between 44321-A2 and the *Streptomyces* as mentioned above. The TYGS database confirmed that 44321-A2 might potentially be a new species and, therefore, named the new strain *Streptomyces actiphen* based on its ability to produce varied actiphenol analogs.}

General NMR and LC-HRMS/MS Materials and Methods. Nuclear magnetic resonance (NMR) spectra were collected using a Bruker 600 NMR spectrometer (^1H : 600 MHz, ^{13}C : 150 MHz) equipped with a Magnex 600/54 active shielded premium magnet, a Bruker liquid N₂ cooled Prodigy cryoprobe, and a Bruker NEO600 console, or a Bruker 800 NMR (^1H : 800 MHz, ^{13}C : 200 MHz) equipped with an Ascend magnet with active shield, a 5 mm triple resonance inverse detection TCI cryoprobe, and a Bruker NEO console. All NMR data analyses were performed using MestReNova NMR software. All chemical shifts were referenced to residual solvent peaks [^1H (DMSO-d₆): 2.50 ppm; ^{13}C (DMSO-d₆): 39.51 ppm].}

LC-HRMS/MS analyses of Biotage fractions, HPLC fractions, and purified compounds were performed using an Agilent 1290 Infinity II UPLC coupled to an Agilent 6545 ESI-Q-TOF-MS system operating in both positive and negative modes. Chromatography was performed using a Phenomenex Kinetex 1.7 μm Phenyl-Hexyl 100 Å (2.1 \times 50 mm) column. The injection volume was 2 μL per sample. The samples were eluted utilizing a gradient starting with a 1 min isocratic wash step consisting of 90% A (95% H₂O/5% MeCN with 0.1% formic acid) and 10% B (100% MeCN with 0.1% formic acid), then a 6 min linear gradient step starting from 10% B to 100% B and ending with 2 min of 100% B wash with a flow rate of 0.4 mL/min. The divert valve was set to MS for 0–7.4 min and set to waste from 7.4 to 9 min. The conditions of the dual AJS ESI were set with the gas temperature at 320 °C, sheath gas temperature at 350 °C, sheath gas flow rate at 11 L/min, and source capillary voltage at 3500 V. The mass range of MS was set to 100–2000 m/z , and the acquisition rate was set to 10 spectra per second. The mass range of MS/MS was set to 50–2000 m/z ; the acquisition rate was set to 6 spectra per second, and the isolation width was set to \sim 1.3 m/z . The collision energy was set based on the following formula: collision energy = $(5 \times m/z)/100 + 10$. The maximum precursor per cycle was set to 9, and the MS/MS mass error tolerance was \pm 20 ppm. The reference masses for positive mode are purine C₅H₄N₄ [M + H]⁺ ion (m/z 121.050873) and hexakis(1H,1H,3H-terfluoropropoxy)phosphazine C₁₈H₁₈F₂₄N₃O₆P₃ [M + H]⁺ ion (m/z 922.009798). The reference masses for negative mode are trifluoroacetic acid (TFA) C₂HF₃O₂ [M – H][–] (m/z 112.985587) and hexakis(1H,1H,3H-terfluoropropoxy)phosphazine C₁₈H₁₈F₂₄N₃O₆P₃ [M + TFA – H][–] (m/z 1033.988109). All solvents used for Biotage fractionation were ACS grade, and those used for HPLC purification and LC-HRMS/MS analyses were HPLC grade or better unless otherwise stated. All LC-MS/MS chromatograms, extracted base peak chromatograms (BPCs), and UV traces at 254 nm in this work were subtracted from the chromatograms of the methanol (MeOH) blank. Data visualization and plotting were performed using GraphPad Prism version 9.4.1 for Mac OS X (GraphPad Software).

Fermentation of *Streptomyces actiphen*. *Streptomyces actiphen* 44321-A2 was streaked onto R2YE agar containing 5 g of yeast extract, 103 gram of sucrose, 10g of dextrose, 0.1g of casamino acid, 0.25g of K₂SO₄, 10.12g of MgCl₂·6H₂O, 5.73 g of TES buffer, 2 mL of trace element solution (containing 10 mg of (NH₄)₆Mo₇O₂₄·4H₂O}, 10 mg of Na₂B₄O₇·10H₂O}, 10 mg of MnCl₂·4H₂O, 10 mg of CuCl₂·2H₂O, 200 mg of FeCl₂·6H₂O, 40 mg of ZnCl₂, and 1L of deionized water, filter sterilized), 10 mL of 0.5% KH₂PO₄, 4 mL of 5 M CaCl₂·2H₂O, 15 mL of 20% L-proline, 7 mL of 1 N NaOH, 25 µg/mL nalidixic acid, 10 µg/mL benomyl, 15 g of agar, and 1 L of double-distilled water. Plates were incubated for 5–7 days at 28 °C. For the strain, 3 mL seed cultures in 14 mL dual-position cap tubes were inoculated with a loopful of vegetative cells from R2YE plates and incubated for 5 days at 28 °C, 200 rpm; 3 mL seed cultures were then inoculated into 100 mL seed cultures in 250 mL baffled flasks and incubated for 7 days at 28 °C, 200 rpm; 50 mL of seed cultures were inoculated into 1 L of fermentation media in 2.8 L baffled Fernbach flasks and grown for 7 days at at 28 °C, 200 rpm.

AT day 7 of the fermentation, 25 g of Amberlite XAD16 resin contained within a polypropylene mesh bag was added to each fermentation culture and agitated overnight at at 28 °C, 200 rpm. On day 8, all resin bags were removed and thoroughly washed with deionized water to remove any water-soluble media components and residual cell mass adsorbed on the resin bags. Each washed resin bag was extracted with 250 mL of MeOH and 250 mL of ethyl acetate (EtOAc). The combined organic fractions were dried under vacuum and redissolved in a minimal amount of MeOH. The solutions were then centrifuged, and the supernatants were loaded onto C18 resin and dried under vacuum prior to Biotage C18 fractionation.

Purification of 2-aminobenzamide-actiphenol. Preparative RF-HPLC fractionation was performed using the same Shimadzu LC-20AP system equipped with a reverse-phase Phenomenex Kinetex[®] 5 mm C18 100 Å (250 × 21.2 mm) column. The materials of strain *Streptomyces actiphen* (brought up in methanol at approximately ~100 to 200 mg/mL) were eluted with a flow rate of 20 mL/min and a linear gradient starting with a 2 min isocratic wash step using 10% acetonitrile/H₂O (with 0.01% TFA), then a 30 min linear gradient step from 10% acetonitrile/H₂O (with 0.01% TFA) to 100% acetonitrile/H₂O (with 0.01% TFA), and then a 5 min wash with 100% acetonitrile/H₂O (with 0.01% TFA) followed by an 8 min equilibration with 10% acetonitrile/ H₂O (with 0.01% TFA). Less than 0.05 mg of the compound is isolated per liter of broth.

Preparative HPLC (Prep HPLC) HF16–18 was fractionated using 40 minutes of an isocratic method with 33% acetonitrile solvent B in H₂O (both solvents A/B with 0.01% TFA) elution. The following compounds were purified using the same semi-preparative system with a reverse-phase Phenomenex Luna[®] 5 mm Phenylhexyl 100 Å (250 × 10 mm) column. 2-Aminobenzamide-actiphenol (approximately 0.05 mg/L) was purified from HF16–18_F24–28 using 30 minutes of an isocratic 35% acetonitrile/H₂O (with 0.01% TFA) elution.

Compound Entries. 2-Aminobenzamide-actiphenol: yellow solid; yellow solid; UV (MeOH) λ_{max} (log ε) 218 (8.9), 267 (6.1), 355 (3.35) nm; ¹H and ¹³C NMR data (Supplementary Table 3); HRESIMS ion peak of C₂₂H₂₂N₃O₅ [M-H]⁻ (*m/z*: found 408.1455, calcd 408.1559).

Cell-Based Bioactivity Analysis. HCT15 (CCI-225) and SW-48 (CCL-231) cells were purchased from ATCC. All cell lines were Mycoplasma free and independently authenticated by short tandem repeat profiling, performed by ATCC. Cells were grown and cultured according to ATCC recommendations. HCT15 cells were cultured in RPMI1640 (30-2001) supplemented with 10% FBS (30-2020). SW48 cells were cultured in Leibovitz's L-15 medium (30-2008) containing 10% FBS. SW-48 cells were grown and treated in an incubator set for atmospheric conditions (no supplemental CO₂ addition). For cell-based

assays, cells were expanded and frozen into single-use aliquots. For each assay, cells were thawed at °C for 1 minute and then immediately re-suspended in 10ml of complete growth medium. Cells were then spun down at 300Xg for 5 minutes and then resuspended in cell-specific growth medium and plated at 2,500 cells per well into Greiner 781080 white cell culture 384-well plates with a total volume per well at 40µl. Natural product extracts or fractions were dissolved in DMSO at 15mg/ml and delivered into the assay plates using Echo 655 acoustic liquid handler instrumentation (Beckman Coulter). Extract and fraction testing concentrations were at 0.25%. For primary screening assays, extract testing was performed n=1 at 0.25% final extract testing concentration (where the original fraction is defined at 100%). Validation assay and fraction studies were performed in triplicate at similar testing concentrations. Negative controls (medium only plus matching 0.25% DMSO) were included in columns 1 and 2. The positive control for these studies was a 10 µM treatment with staurosporine in columns 23 and 24 of each assay plate. Samples were interrogated in wells A03 to P22. The high-throughput data software Mscreen was utilized for primary hit, validation selection, and analysis of concentration-response curve results [<https://doi.org/10.1177/1087057112450186>]. Following compound addition, cells were cultured for 48 hours at either 5% CO₂ at 37°C for HCT15 cells or atmospheric air at 37°C for SW-48 cells. Cell viability was measured using a CellTiter-Glo luminescent kit (catalog no. G7571) from Promega as directed using a PHERAstar instrument from BMG Labtech.

Below sentence is added to the discussion section to conclude this finding:

Actiphenol, a prominent member of the glutarimide-containing polyketide family, has long been recognized for its role as a eukaryotic translation inhibitor. Seq2PKS unveiled a new variant of actiphenol, named 2-aminobenzamide-actiphenol from *Streptomyces actiphen*. This molecule diverges notably from classical actiphenol structures by lacking a methyl group at C-4 and introducing both a new methylene group and an aminobenzamide moiety. Although initial tests revealed promising cytotoxic effects in crude extracts against colon cancer cell lines, the isolated variant did not exhibit substantial activity in further assays. Further studies are needed to explore the bioactive properties of this new compound.

C2.6 In the comparative analysis with existing methods such as Prism and antiSMASH, please explain how can get the advantages by Seq2PKS, if possible that the underlying reasons should be elucidated with data to support the Seq2PKS's enhancements in key metrics.

R2.6. Seq2PKS improves on current existing programs PRISM and antiSMASH in a couple of aspects. In the case of PRISM, we only have access to overall accuracy, but for other methods, we have highlighted step-by-step accuracy benchmarks. Overall, Seq2PKS achieves 94.4% accuracy, which is comparable to antiSMASH in AT-domain specificity prediction. However, Seq2PKS improves prediction accuracy for AT-domain specificity by using an extra-tree-based model optimized to achieve high accuracy for test data points that are far from training data. To measure distance between test data points and training data points, we used Hamming distance on the 24 amino acid signatures [[https://doi.org/10.1016/S0022-2836\(03\)00232-8](https://doi.org/10.1016/S0022-2836(03)00232-8)]. For testing samples that are at least six Hamming distances away from any training data points, Seq2PKS achieve accuracy of 51%, compared to 38% for antiSMASH (Figure 2(a)). During the assembly line order prediction process, Seq2PKS employs a logistic regression algorithm, taking the docking domain sequences as input to predict the optimal assembly pathway for candidate core structures. This method achieves 92% accuracy for non-co-linear polyketide BGCs, while DDAP and antiSMASH achieve 55% and 74% accuracy, respectively. Seq2PKS also improves on existing methods by applying more comprehensive post-assembly modifications. Overall, Seq2PKS predicts 8 out of 80 polyketides in the benchmark dataset from the PRISM paper correctly (16 if counting polyketides outside PRISM dataset), while PRISM and antiSMASH do not predict any of the

polyketides correctly. The highest Tanimoto similarity that PRISM and antiSMASH achieve are 0.65 and 0.30.

Figure 2(a): The accuracy of different classifiers for predicting the substrate specificity of AT-domains. The notation B_k+ represents the bin containing data points at least k Hamming distances away from any training data points. While most methods achieve similar accuracy, the extra tree classifier generalizes better, i.e., it achieves higher accuracy for testing samples that are dissimilar to the training samples.

We added the following sentence to the discussion:

Compared to other methods, in AT-domain specificity prediction, Seq2PKS achieves over 94% accuracy, comparable to the accuracy for antiSMASH. However, Seq2PKS improves prediction accuracy for AT-domain specificity by using an extra-tree-based model optimized to achieve high accuracy for test data points that are far from training data. For testing samples that are at least six Hamming distances away from any training data points, Seq2PKS achieves an accuracy of 51%, compared to 38% for antiSMASH. During the assembly line order prediction process, Seq2PKS employs a logistic regression algorithm, taking the docking domain sequences as input to predict the optimal assembly pathway for candidate core structures. This method achieves 92% accuracy for non-co-linear polyketide BGCs, while DDAP and antiSMASH achieve 55% and 74% accuracy, respectively. Furthermore, Seq2PKS uses a subgraph isomorphism algorithm to consider a more comprehensive set of post-assembly modifications. Overall, Seq2PKS correctly predicts eight out of 80 polyketides in the benchmark dataset (16 if counting polyketides without PRISM result), while PRISM and antiSMASH do not predict any of the polyketides correctly. The highest Tanimoto similarity that PRISM and antiSMASH achieve are 0.65 and 0.30.

C 2.7. A more detailed ReadMe file should be given for the user to access/read the codes.

R2.7 We have addressed the reviewer's comment, please see our response R1.5 above. The server is now accessible at <https://run.npanalysis.org/>. User can use the server to avoid running manually.

We also provide a more detailed explanation about the code in which folder is responsible for which task (detailed in GitHub ReadMe at <https://github.com/mohimanilab/Seq2PKS>). A sample test is also provided in the ReadMe, see below:

Docker(Recommended)

Create your working directory,

```
mkdir Seq2PKS
cd Seq2PKS
```

Interacting with the container,

```
sudo docker run -it -v $(pwd):/usr/src/app/mnt --privileged --entrypoint /bin/bash seq2pks
```

Here is an example of running with a FASTA file and a MZ file as input:

```
cp /path/to/source/{sample.fasta,sample_spectra.mzML} .
python main.py --sequence_file sample.fasta --sequence_id sample --pattern dereplicator_plus -
```

We have the sample run result included in the "test_result" folder for testing purposes. You should be able to generate the exact same result in the folder by following the below command,

```
python main.py --ncbi_id DQ149987.1 --pattern dereplicator_plus --spectrum_path sample_spectra
cp -r test_result mnt
```

The identified compounds are stored in the file

"test_result/DQ149987/compound/backbone2pks_result/database.csv". The generated compounds are scored against the input spectrum file using dereplicator+, the obtained match is stored in "test_result/DQ149987/compound/backbone2pks_result/psms.tsv".

Reviewer #3 (Remarks to the Author):

C3.1 In their paper, Yan et al. present a new computational pipeline for improving the prediction of the structures of compounds derived from polyketide synthases (PKS) and their direct association with mass spectrometry (MS) data. This pipeline has the potential to enhance the automated structure prediction of natural products originating from biosynthetic gene clusters (BGCs).

However, the manuscript's current readability may not cater to a broad audience, especially within the context of Nature Communications.

R3.1 Thank you for the suggestions on our paper. We have provided a point-by-point response to each of your comments below. The changes are indicated in red in the revised manuscript and the Supplement.

C3.2 The manuscript requires a more substantial background in both biosynthetic logic and the computational aspects of PKS systems for a comprehensive understanding. The Results and Methods sections are concise and would benefit from further clarification regarding the rationale behind specific steps and the advantages they confer.

R3.2 We greatly appreciate the reviewer's comment. We have revised the introduction and method sections to further explain each step of Seq2PKS and the rationale behind them.

Introduction section:

A modular Type I Cis-AT polyketide synthase (MT1PKS) consists of various modules, each responsible for synthesizing a specific chemical substructure in the core structure of an MT1PK. Each module comprises an acyl transferase (AT) domain responsible for recruiting initial substrates to an acyl carrier domain (ACP), and other domains within each module can further modify this substrate to form the mature substructure. Substructures attached to ACPs of neighboring modules are connected by keto synthase (KS) domains in a modular assembly pathway, where starting from the substructure produced by the first module, each subsequent module extends the polyketide chain by adding a new substructure and passing it to the next module as ACP-tethered intermediates. The last module contains a thioesterase (TE) domain that releases the polyketide core structure from the MT1PKS as a cyclic or linear product. Tailoring enzymes then catalyze post-assembly modifications of the MT1PKS product into a mature bioactive molecule [<https://doi.org/10.1021/bi500290t>, [https://doi.org/10.1016/S0076-6879\(09\)04601-1](https://doi.org/10.1016/S0076-6879(09)04601-1)].

Based on this biosynthetic logic, predicting the chemical structure of polyketides from their BGCs is a challenging task that involves (a) annotating PKS domains and enzymes, (b) predicting substrates from AT domains and other domains present in each module, (c) predicting the order of substrates in the polyketide assembly pathway, and (d) applying corresponding post-assembly modifications to the polyketide core structure based on identified modification genes.

Method section:

(b) Predicting Substrates. Previous studies have shown that the substrate specificity of AT domains is largely determined by specific amino acid residues of the AT active site pocket [<https://doi.org/10.1046/j.1365-2958.2002.02815.x>]. Therefore, for each AT-domain, Seq2PKS extracts 24 active site residues reported by Yadav et al. [[https://doi.org/10.1016/S0022-2836\(03\)00232-8](https://doi.org/10.1016/S0022-2836(03)00232-8)] by aligning them to the reference sequences using MUSCLE sequence alignment [<https://doi.org/10.1093/nar/gkh340>]. These 24 active site residues were identified by analyzing the crystal structure of *E. coli* FAS AT domain [<https://doi.org/10.1074/jbc.270.22.12961>] and have proven to show distinct patterns across different AT domain specificities. Seq2PKS represents these signatures as one-hot encoded vectors, where each amino acid maps to a unique twenty-dimensional binary vector containing nineteen zeros and a single one. An extra tree-based model is trained to predict the domain specificity based on these vector representations. The number of trees in the forest is set to 150 and the max depth of the tree is set to 10 to achieve optimal performance.

Following the initial substrate identification from an AT domain, other domains within the same module including ketoreductase (KR), dehydratase (DH), and enoylreductase (ER) domains can modify the substrate further. Specifically, the KR domain catalyzes the reduction of the substrate carbonyl group to a hydroxy group. A DH domain then removes a water molecule from the hydroxy substructure, leading to the formation of an olefin substructure. Finally, an ER domain reduces the double bond introduced by the DH domain, resulting in saturation of the polyketide chain. This sequence of modifications contributes to the remarkable structural diversity and complexity of polyketide compounds. Based on this logic, Seq2PKS recruits a rule-based approach (Supplementary Figure 3) that predicts each substructure, i.e. modified substrate, using the substrate specificity of the AT domain and the presence of substrate-modifying domains in the module).

(c) Predicting the assembly pathway. A single helical segment from the C-terminal linker of one MT1PKS open reading frame (ORF) and three helical segments from the N-terminal linker of the subsequent MT1PKS ORF represent structures known as docking domains which direct the interactions of neighboring MT1PKS proteins in the assembly pathway [[https://doi.org/10.1016/S1074-5521\(03\)00156-X](https://doi.org/10.1016/S1074-5521(03)00156-X)]. An ORF can follow another ORF in the assembly line only if the docking domain at the C-terminus of the former MT1PKS ORF can interact with the N-terminal docking domain of the latter MT1PKS ORF. Therefore, interactions between docking domains can be used to infer the order of ORFs in the assembly line. Seq2PKS calculates a docking interaction score between terminal modules in a BGC. Each module's head (i.e. N-terminal) and tail (i.e. C-terminal) docking domains are extracted as the first 100 amino acid residues from the C-terminus and the first 50 amino acid residues from the N-terminus, respectively. The head domain from the first module and the tail domain from the last module of the MT1PKS pathway are excluded.

(d) Incorporating post-assembly modifications. Once the polyketide core structures are synthesized, they undergo various post-assembly modifications. These modifications are chemical alterations driven by tailoring enzymes that affect the properties and functionalities of the polyketides. Key modifications include the addition or removal of phosphate groups (phosphorylation and dephosphorylation) [<https://doi.org/10.1039/B801658P>], the attachment of acetyl groups (acetylation) [<https://doi.org/10.3390/molecules16076092>], and the incorporation of sugar molecules (glycosylation) [<https://doi.org/10.1038/s41589-019-0314-6>]. Seq2PKS predicts hypothetical structures of mature polyketides by applying various combinations of modifications corresponding to the identified tailoring enzymes. To this end, we extracted polyketide tailoring enzymes and their corresponding modifications by literature mining and parsed them in a computer-readable format (Figure 13). For each modification, the reaction motif is stored as a SMILES string, along with a series of graph modifications (addition/removal of nodes/edges) that are applied to the motif. Figure 5 summarizes all the modifications extracted from known polyketide molecules.

C 3.3 Additionally, the discussion section could be improved by comparing the proposed tool with existing tools like NPlinker, which also aim to link BGCs with their respective natural products.

R 3.3 We have added the comparison between Seq2PKS with existing MS data matching tools like NPlinker to the discussion section:

Currently, two types of tools bridge metabolomics and genomic data. Correlation-based approaches like NPlinker [<https://doi.org/10.1371/journal.pcbi.1008920>], NPOmix [<https://doi.org/10.1093/pnasnexus/pgac257>], and other metabolomics methods [<https://doi.org/10.1038/nchembio.1659>, <https://doi.org/10.1038/s41589-019-0400-9>, <https://doi.org/10.1038/s41589-023-01276-8>] focus on linking gene cluster families (GCFs) with molecular

families (MFs) or spectra, based on co-occurrence of molecular features and BGCs. Conversely, feature-based approaches like GNP [<https://doi.org/10.1038/ncomms9421>], MetaMiner [<https://doi.org/10.1016/j.cels.2019.09.004>], Seq2RiPP [<https://doi.org/10.1038/s41467-023-39905-4>] and NRPminer [<https://doi.org/10.1038/s41467-021-23502-4>] strive to associate BGCs with mass spectra by predicting the hypothetical structure of BGC products, followed by in silico mass spectral database search. Seq2PKS is a feature-based method that integrates metabolomics and genomics data capabilities for the efficient and scalable identification of polyketides.

C 3.4 The manuscript does not provide clear information about the overlap between the training and test datasets used in the pipeline. It is essential to ensure there is no such overlap to ensure the reliability of the results.

R 3.4 Thanks for the comment. To avoid overlap between training and test data, all the models involved in the manuscript are evaluated by five-fold cross-validation. First, the evaluation of accuracy for AT domain specificity prediction is derived from five-fold cross-validation. We have updated the following paragraph regarding the cross-validation step for the machine learning models:

To compare different machine learning algorithms for Cis-AT domain specificity, we conducted five-fold cross-validation using the 624 labeled training data and reported the average resulting accuracy (Figure 2). While most algorithms exhibit similar accuracy in the overall category, the extra-tree algorithm demonstrates better generalization (i.e. it attains higher accuracies for bins that are far away from training domains based on Hamming distance).

The evaluation of accuracy for predicting assembly order is also done using five-fold cross-validation:

Figure 4 illustrates the five-fold cross-validation accuracy of the PNN method compared with the existing method DDAP, which employs a support vector machine algorithm [<https://doi.org/10.1093/bioinformatics/btz677>]. As a benchmark dataset, we used docking domain sequences from 90 BGCs reported in the DDAP paper.

We also added the below sentence to the method section:

Cross-validation for machine learning model. During the substrate specificity prediction and the assembly line order prediction process, standard machine learning models are employed. These models include Logistic Regression, Support Vector Machine, K Nearest Neighbor, Multilayer Perceptron, Random Forest, Decision Tree, Bernoulli Naive Bayes, Gaussian Naive Bayes, and Extremely Randomized Trees. To ensure there is no sample overlap, five-fold cross-validation is utilized, where the datasets are randomly split into five subsets with the same number of samples. In each step, four of the subsets are used for training, and the remaining one serves as test data.

Five-fold cross-validation offers a more accurate estimation of model performance, particularly when the training data is limited. This process is executed five times for each selected model, with the dataset being randomly shuffled each time. The accuracies are calculated by averaging across these cross-validations (Figure 2 and Figure 4).

C 3.5 While the identification of putative gene clusters associated with the presented molecules is promising, the lack of experimental validation leaves uncertainty about whether these identified BGCs indeed encode the suggested compounds.

R3.5 We thank the reviewer for bringing this up. We used two methods to validate the products. Due to the unavailability of the oasomycin A standard, our analysis primarily focused on the validation of the monazomycin BGC. First, we cultured *Streptomyces cinnamoneus* NRRL B-24434 in ISP2 medium. This was followed by extraction of the fermentation broth with 1-butanol, rotary evaporation of the 1-butanol phase, and resuspension of the extract in methanol. We then performed LC-MS (Liquid Chromatography-Mass Spectrometry) analysis on the 1-butanol extract. The LC-MS results, compared with an authentic monazomycin standard from Santa Cruz Biotechnology, are depicted in the figure below (Supplementary Figure 11,12). The retention time and mass spectrum observed from our samples matched that of the authentic standard, confirming that the strain indeed produces monazomycin as expected. We also annotated the BGC of monazomycin to ensure that the machinery for the production of the polyketides is present, as shown below (Supplementary Tables 1). We have added the following sentence to the results section:

Experimental validation of monazomycin BGC. To further corroborate the novel BGC identified for monazomycin, we extensively annotated genes in the BGC (Supplementary Table 1). The annotation results confirm this BGC has the necessary biosynthetic machinery to produce monazomycin.

We further cultivated the strain *Streptomyces cinnamoneus* NRRL B-24434 and collected LC-MS/MS data on the butanolic extracts of its growth medium. Comparison with the authentic standard in both retention time (Supplementary Figure 11) and tandem mass spectrometry data (Supplementary Figure 12) was consistent with monazomycin production by *Streptomyces cinnamoneus* NRRL B-24434.

Supplementary Table 1: Gene annotation for identified monazomycin BGC in *Streptomyces cinnamoneus* NRRL B-24434.

	Location	Nucleotide	Amino Acid	Hypothesized Function
Contig 1	2 - 1,417	1,416	472	LuxR family transcriptional regulator
Contig 2	1,573 - 2,337	765	254	alpha/beta fold hydrolase (thioesterase)
Contig 3	3,173 - 5,959	2,787	928	LuxR family transcriptional regulator
Contig 4	6,241 - 7,899	1,659	552	NAD(P)/FAD-dependent oxidoreductase
Contig 5	8,450 - 9,154	705	234	thioesterase
Contig 6	9,280 - 10,692	1,413	470	class I adenylate-forming enzyme family
Contig 7	10,751 - 11,701	951	316	ACP S-malonyltransferase
Contig 8	11,733 - 13,025	1,293	430	histidine kinase
Contig 9	13,359 - 14,900	1,542	513	glycosyltransferase
Contig 10	14,973 - 16,214	1,242	413	cytochrome P450
Contig 11	16,343 - 19,726	3,384	1,127	SDR family NAD(P)-dependent oxidoreductase
Contig 12	20,001 - 29,660	9,660	3,219	type I polyketide synthase
Contig 13	29,748 - 45,353	15,606	5,201	type I polyketide synthase
Contig 14	45,420 - 56,489	11,070	3,689	type I polyketide synthase
Contig 15	56,536 - 71,967	15,432	5,143	type I polyketide synthase
Contig 16	72,473 - 86,383	13,911	4,636	type I polyketide synthase

Contig 17	86,428 - 107,196	20,769	6,922	type I polyketide synthase
	107,219 -			
Contig 18	125,716	18,498	6,165	type I polyketide synthase
	126,193 -			
Contig 19	138,252	12,060	4,019	type I polyketide synthase
	138,360 -			
Contig 20	138,524	165	55	unknown

Supplementary Figure 11: LCMS-based comparison of monazomycin analyte from our cultured strain *Streptomyces cinnamoneus* NRRL B-24434 and an authentic standard provided by Santa Cruz Biotechnology.

Supplementary Figure 12: Tandem mass spectra for monazomycin from our cultured strain *Streptomyces cinnamoneus* NRRL B-24434 (top) and authentic standard (bottom) provided by Santa Cruz Biotechnology.

Additionally, we have incorporated the following sentence into the methods section:

Validation of monazomycin BGC. We cultivated liquid cultures of *Streptomyces cinnamoneus* NRRL B-24434 in ISP2 medium for 7 days at 28°C. The shaking speed is 220 rpm with a flask size of 1L and medium content of 100mL. This was followed by extraction of the broth with 1-butanol, rotary evaporation of the 1-butanol extract, resuspension of the extract in 80% methanol, followed by LC-MS/MS analysis. LC-MS data was collected on a Thermo QExactive Orbitrap connected to a Vanquish LC system. LC settings were as follows: injection volume 5 µl; Phenomenex Kinetex 2.6 µm C18 reverse phase 100 Å 150 mm x 3 mm LC column; LC gradient, solvent A, 0.1% formic acid; solvent B, acetonitrile (0.1% formic acid); 0 min, 10% B; 5 min, 60% B; 5.1 min, 95% B; 6 min, 95% B; 6.1 min, 10% B; 9.9 min, 10% B; 0.5 ml/min. MS settings were as follows: positive ion mode; full MS, resolution 70,000; mass range 400-1,200

m/z; dd-MS2 (data-dependent MS/MS), resolution 17,500; AGC target 1×10^5 , loop count 5, isolation width 1.0 m/z, collision energy 25 eV, dynamic exclusion 0.5 s.

In addition, we identified a novel actiphenol variant 2-aminobenzamide-actiphenol using Seq2PKS. The finding was validated with extensive experiments. This finding further proves the power of Seq2PKS in discovering novel compounds. We have added below sentence in the result section to address this novel finding.

Identification of novel actiphenol variant 2-aminobenzamide-actiphenol. We applied Seq2PKS to the *Streptomyces actiphen* genome available through the University of Michigan Natural Product Discovery Core and identified a long PK system with 55% similarity to the previously proposed actiphenol BGC. Seq2PKS generated one core structure, resulting in 258 mature compounds for this PK BGC. By searching the constructed molecules in a spectral dataset using Dereplicator+, we identified two PK-spectral matches with scores of 21 and 28. We extensively annotated the BGC for these molecules (Supplementary Table 2) and compared it with the reported BGC for actiphenol from MiBIG database (Supplementary Figure 13). The former molecule is a previously reported variant of actiphenol called Nong-Kang 101-G (Figure 9) [http://sioc-journal.cn/Jwk_hxxb/EN/abstract/article_342462.shtml], while the latter is a novel actiphenol congener in which the cyclohexanone unit is substituted by a phenol moiety (Figure 10) [<https://doi.org/10.1038/nchembio.304>].

The new actiphenol molecule was isolated as a yellow solid with a molecular formula of $C_{22}H_{22}N_3O_5$, derived from an HRESIMS ion peak of $C_{22}H_{22}N_3O_5 [M-H]^-$ (m/z : found 408.1455, calcd 408.1559; NMR data for the new molecule is shown in Supplementary Figures 14,15,16,17,18,19 and Supplementary Table 3). Careful analyses of the 1D and 2D NMR spectra, confirmed the actiphenol moiety, however, the C-4 methyl group is absent. Instead, we observed a new methylene ($\delta_H = 4.30$) at C-15 ($\delta_C = 45.9$) that correlates with C-3 ($\delta_C = 136.9$), C-5 ($\delta_C = 127.6$), and an aromatic carbon C-16 (chemical shift of ($\delta_C = 149.9$)) atypical of the actiphenol spin systems. Further examination revealed the presence of an additional aminobenzamide moiety attached to the actiphenol through an N-C bond. The HMBC correlations between the triplet H-18 ($\delta_H = 7.23$)/C-16 ($\delta_C = 149.9$) and the doublet H-20 ($\delta_H = 7.61$)/C-16 ($\delta_C = 149.9$) and C-22 ($\delta_C = 171.9$) confirmed the relative position of the secondary amine ($\delta_H = 7.23$) and the primary amide on this aromatic ring. Based on its structure, we named this molecule 2-aminobenzamide-actiphenol (Figure 11).

We tested the molecules against two cancer cell lines, SW48 (CCL231, colon cancer) and HCT15 (CCL-225 colon cancer) based on the initial strong cytotoxic activity observed (Supplementary Figure 20) from the crude extracts generated by the producing strain (*Streptomyces actiphen*) during the initial isolation. However, the isolated novel molecule 2-aminobenzamide-actiphenol did not show appreciable activity upon isolation and characterization (Supplementary Figure 21)

Figure 9: Identification of Nong-Kang 101-G by Seq2PKS. (a) Domains and modules in the biosynthetic genes are identified. (b) Substrate specificity and mature substrate for each module are predicted. (c) Assembly order is predicted, and the core structure is constructed by connecting the predicted mature substrates for each module. (d) Cyclization and hydroxylation modifications are applied by two enzymes from the BGCs (shown in pink and orange, respectively). (e) Fragments from the hypothetical molecule (shown in black) that match a peak in mass spectra are highlighted (shown in red). Fragments are generated by one or two rounds of fragmentation of the hypothetical molecule. (f) The annotated spectrum of the hypothetical molecule is shown in red.

Figure 10: Identification of 2-aminobenzamide-actiphenol by Seq2PKS. (a) Domains and modules in the biosynthetic genes are identified. (b) Substrate specificity and mature substrate for each module are predicted. (c) Assembly order is predicted, and the core structure is constructed by connecting the predicted mature substrates for each module. (d) Cyclization and dehydroxylation modifications are applied by two enzymes from the BGCs (shown in pink and orange, respectively). While Seq2PKS cannot predict the presence of 2-aminobenzamide, it correctly predicts its molecular formula from the difference between the precursor mass spectrum and the total mass of the hypothetical molecule. (e) Fragments from the hypothetical molecule (shown in black) that match a peak in mass spectra are highlighted (shown in red). Fragments are generated by one or two rounds of fragmentation of the hypothetical molecule. (f) The annotated spectrum of the hypothetical molecule is shown in red.

Chemical Formula: $C_{22}H_{23}N_3O_5$

Exact Mass: 409.16

Figure 11: 1H NMR Spectral Analysis showcasing the chemical shifts and multiplicity patterns for the identification of 2-aminobenzamide-actiphenol.

Supplementary Table 2: Gene annotation for 2-aminobenzamide-actiphenol BGC.

	Location	Nucleotide	Amino Acid	Hypothesized Function
Contig 1	2 - 901	900	299	Polar amino acid ABC transporter
Contig 2	905 - 1,657	753	250	ABC transporter
Contig 3	1,753 - 2,538	786	261	Unknown
Contig 4	2,608 - 3,297	690	229	GntR family transcriptional regulator
Contig 5	3,343 - 4,149	807	268	Short-chain dehydrogenase/reductase SDR
Contig 6	4,370 - 5,017	648	215	Unknown
Contig 7	5,017 - 5,667	651	216	Unknown
Contig 8	5,664 - 6,539	876	291	ABC transporter
Contig 9	6,536 - 7,639	1,104	367	Transport system permease protein
Contig 10	7,620 - 8,774	1,155	384	Iron compound ABC transporter
Contig 11	9,012 - 9,158	147	48	Unknown
Contig 12	9,460 - 10,227	768	255	Unknown
Contig 13	10,475 - 11,200	726	241	Unknown
Contig 14	11,658 - 12,803	1,146	381	Unknown
Contig 15	12,858 - 13,307	450	149	Unknown
Contig 16	14,117 - 14,467	351	116	Unknown
Contig 17	14,738 - 15,448	711	236	Unknown
Contig 18	16,234 - 16,437	204	67	Unknown
Contig 19	16,458 - 17,060	603	200	Unknown
Contig 20	17,162 - 17,626	465	154	Unknown
Contig 21	17,662 - 18,789	1,128	375	Serine/threonine protein kinase
Contig 22	18,966 - 19,784	819	272	Unknown
Contig 23	20,001 - 40,511	20,511	6,836	Type I polyketide synthase
Contig 24	40,604 - 42,643	2,040	679	Asparagine synthase
Contig 25	42,660 - 42,917	258	85	Putative acyl carrier protein
Contig 26	42,921 - 43,796	876	291	Type I polyketide synthase
Contig 27	44,081 - 47,410	3,330	1,109	Transcriptional regulator
Contig 28	47,582 - 50,980	3,399	1,132	NRP synthase
Contig 29	51,124 - 52,323	1,200	399	Cytochrome P450
Contig 30	52,477 - 53,262	786	261	Short-chain dehydrogenase/reductase SDR
Contig 31	53,900 - 54,988	1,089	362	Flavin oxidoreductase
Contig 32	55,091 - 56,020	930	309	LysR family transcriptional regulator

Supplementary Table 3: NMR result for 2-aminobenzamide-actiphenol.

Position	δ_c (type)	δ_H , multiplets (J in Hz)	COSY	HMBC	ROESY
1	159.3 (C)				
1-OH		12.33, s		1, 2, 6	
2	127.0 (C)				
3	136.9 (CH)	7.45, s		1, 5, 14, 15	14, 15a, 15b
4	130.0 (C)				
5	127.6 (CH)	7.82, s		1, 3, 7, 15	8a, 8b, 15a, 15b
6	118.7 (C)				
7	205.6 (C)				
8a	42.9 (CH ₂)	3.18, d (6.4)	9	7, 9, 10, 13	5, 10a, 10b, 13a, 13b
8b	42.9 (CH ₂)	3.18, d (6.4)	9	7, 9, 10, 13	5, 10a, 10b, 13a, 13b
9	26.6 (CH)	2.62, m	8a, 8b		
10a	37.5 (CH ₂)	2.39, dd (16.5, 10.7)	10b	11, 12	5
10b	37.5 (CH ₂)	2.58, dd (16.5, 10.7)	10a	11, 12	5
11	173.4 (C)				
11-NH		10.76, s		10, 13	
12	173.4 (C)				
13a	37.5 (CH ₂)	2.39, d (16.5, 10.7)	13b	11, 12	5
13b	37.5 (CH ₂)	2.58, d (16.5, 10.7)	13a	11, 12	5
14	15.8 (CH ₃)	2.17, s		1, 2, 3	3
15a	45.9 (CH ₂)	4.30, s		3, 4, 5, 16	3, 5, 17
15b	45.9 (CH ₂)	4.30, s		3, 4, 5, 16	3, 5, 17
15-NH		8.47, br s			
16	149.9 (C)				
17	112.1 (CH)	6.69, d (8.4)	18	19, 21	15a, 15b, 18
18	132.8 (CH)	7.23, t (7.9)	17, 19	16, 20	17, 19
19	114.9 (CH)	6.54, t (7.5)	18, 20	17, 21	18, 20
20	129.5 (CH)	7.61, d (7.8)	19	16, 18, 22	18, 19
21	114.9 (C)				
22	171.9 (C)				
22-NHa		7.17, br s			
22-NHb		7.86, br s			

Supplementary Figure 13: Comparison of Biosynthetic Gene Clusters (BGCs) between the 2-aminobenzamide-actiphenol and actiphenol from the MIBiG database (BGC0000175).

Supplementary Figure 14: ¹H spectrum of 2-aminobenzamide-actiphenol.

Supplementary Figure 15: ^{13}C spectrum of 2-aminobenzamide-actiphenol.

Supplementary Figure 16: COSY spectrum of 2-aminobenzamide-actiphenol.

Supplementary Figure 17: HSQC spectrum of 2-aminobenzamide-actiphenol.

Supplementary Figure 18: ^1H - ^{13}C spectrum of 2-aminobenzamide-actiphenol.

Supplementary Figure 19: ROESY spectrum of 2-aminobenzamide-actiphenol.

44321_A2I-80L-Crude

Supplementary Figure 20: Activity of crude extracts generated from 2-aminobenzamide-actiphenol producing strain *Streptomyces actiphen* against SW48 (CCL231, Colon Cancer) and HCT15 (CCL-225 Colon Cancer) cell line.

2-aminobenzamide-actiphenol (20)

Supplementary Figure 21: Activity of the isolated novel molecule 2-aminobenzamide-actiphenol against SW48 (CCL231, Colon Cancer) and HCT15 (CCL-225 Colon Cancer) cell line.

We added below part to the method section to discuss the experimental protocol:

Genome Extraction, Sequencing, Assembly, and Annotation. The genomic DNA for 44321-A2 was extracted using BIOSEARCH Technologies MasterPure Complete DNA and RNA purification kit following the manufacturer's protocol with some modifications, including additional lysis steps using EDTA (50mM) and lysozyme (10mg/mL) and one more step of heat treatment at 95 °C before RNase A treatment. The genomic DNA was sequenced at the University of Minnesota, Genomics Center, Minneapolis, MN. The sample library was prepared using PacBio Sequel II HiFi -- the SMRT Cell 8M typically generates approximately 4--5 million raw reads with flexible sequencing run times of up to 30 hours, yielding a 1.04 GB fastq file. Demultiplexing and quality control were done using PATRIC service which obtained a total of 90,490 read pairs. The 44321-A2 was assembled using Canu version 1.7.1 and 2 rounds of polishing done as iteration using Racon version 2.4.13 as a part of the comprehensive genome analysis service at

PATRIC, QUAST version 5.0.2, minimap2 (2.17-r974-dirty), samtools Version 1.11, and Bandage 0.8.1 with default parameters were used for assembly quality assessment and visualization. The resulting assembled genome has an estimated length of 8,799,402 bp and an average GC content of 72.18%. Two contigs generated for this genome have 7,647 protein coding sequences (CDS), 68 transfer RNA (tRNA) genes, and 18 ribosomal RNA (rRNA) genes. The annotation included 2,918 hypothetical proteins and 4,729 proteins with functional assignments. The proteins with functional assignments included 1,270 proteins with Enzyme Commission (EC) numbers, 1,111 with Gene Ontology (GO) assignments, and 1,007 proteins that were mapped to KEGG pathways. Then the assembled contig FASTA file is used to extract the 16S fragments with the help of ContEst16S tool at Ezbiocloud. Whole-genome similarity metrics, including average nucleotide identity (ANI) and DNA–DNA hybridization (DDH), were obtained to estimate genetic relatedness and define phylogeny. FastANI showed 86% genome-relatedness of our microbial strain genome to *Streptomyces atratus* ASM333086v1 and *Streptomyces gelaticus* ASM1464953v1. A whole-genome-sequence-based phylogenetic tree was built using the TYGS analysis method further supporting the genetic closeness between 44321-A2 and the *Streptomyces* as mentioned above. The TYGS database confirmed that 44321-A2 might potentially be a new species and, therefore, named the new strain *Streptomyces actiphen* based on its ability to produce varied actiphenol analogs.}

General NMR and LC-HRMS/MS Materials and Methods. Nuclear magnetic resonance (NMR) spectra were collected using a Bruker 600 NMR spectrometer (^1H : 600 MHz, ^{13}C : 150 MHz) equipped with a Magnex 600/54 active shielded premium magnet, a Bruker liquid N₂ cooled Prodigy cryoprobe, and a Bruker NEO600 console, or a Bruker 800 NMR (^1H : 800 MHz, ^{13}C : 200 MHz) equipped with an Ascend magnet with active shield, a 5 mm triple resonance inverse detection TCI cryoprobe, and a Bruker NEO console. All NMR data analyses were performed using MestReNova NMR software. All chemical shifts were referenced to residual solvent peaks [^1H (DMSO-*d*₆): 2.50 ppm; ^{13}C (DMSO-*d*₆): 39.51 ppm].}

LC-HRMS/MS analyses of Biotage fractions, HPLC fractions, and purified compounds were performed using an Agilent 1290 Infinity II UPLC coupled to an Agilent 6545 ESI-Q-TOF-MS system operating in both positive and negative modes. Chromatography was performed using a Phenomenex Kinetex 1.7 μm Phenyl-Hexyl 100 Å (2.1 \times 50 mm) column. The injection volume was 2 μL per sample. The samples were eluted utilizing a gradient starting with a 1 min isocratic wash step consisting of 90% A (95% H₂O/5% MeCN with 0.1% formic acid) and 10% B (100% MeCN with 0.1% formic acid), then a 6 min linear gradient step starting from 10% B to 100% B and ending with 2 min of 100% B wash with a flow rate of 0.4 mL/min. The divert valve was set to MS for 0–7.4 min and set to waste from 7.4 to 9 min. The conditions of the dual AJS ESI were set with the gas temperature at 320 °C, sheath gas temperature at 350 °C, sheath gas flow rate at 11 L/min, and source capillary voltage at 3500 V. The mass range of MS was set to 100–2000 m/z , and the acquisition rate was set to 10 spectra per second. The mass range of MS/MS was set to 50–2000 m/z ; the acquisition rate was set to 6 spectra per second, and the isolation width was set to \sim 1.3 m/z . The collision energy was set based on the following formula: collision energy = $(5 \times m/z)/100 + 10$. The maximum precursor per cycle was set to 9, and the MS/MS mass error tolerance was \pm 20 ppm. The reference masses for positive mode are purine C₅H₄N₄ [M + H]⁺ ion (m/z 121.050873) and hexakis(1H,1H,3H-terfluoropropoxy)phosphazine C₁₈H₁₈F₂₄N₃O₆P₃ [M + H]⁺ ion (m/z 922.009798). The reference masses for negative mode are trifluoroacetic acid (TFA) C₂HF₃O₂ [M – H][–] (m/z 112.985587) and hexakis(1H,1H,3H-terfluoropropoxy)phosphazine C₁₈H₁₈F₂₄N₃O₆P₃ [M + TFA – H][–] (m/z 1033.988109). All solvents used for Biotage fractionation were ACS grade, and those used for HPLC purification and LC-HRMS/MS analyses were HPLC grade or better unless otherwise stated. All LC-MS/MS chromatograms, extracted base peak chromatograms (BPCs), and UV traces at 254 nm in this work were subtracted from the chromatograms of the methanol (MeOH) blank. Data visualization and plotting were performed using GraphPad Prism version 9.4.1 for Mac OS X (GraphPad Software).

Fermentation of *Streptomyces actiphen*. *Streptomyces actiphen* 44321-A2 was streaked onto R2YE agar containing 5 g of yeast extract, 103 gram of sucrose, 10g of dextrose, 0.1g of casamino acid, 0.25g of K₂SO₄, 10.12g of MgCl₂·6H₂O, 5.73 g of TES buffer, 2 mL of trace element solution (containing 10 mg of (NH₄)₆Mo₇O₂₄·4H₂O}, 10 mg of Na₂B₄O₇·10H₂O}, 10 mg of MnCl₂·4H₂O, 10 mg of CuCl₂·2H₂O, 200 mg of FeCl₂·6H₂O, 40 mg of ZnCl₂, and 1L of deionized water, filter sterilized), 10 mL of 0.5% KH₂PO₄, 4 mL of 5 M CaCl₂·2H₂O, 15 mL of 20% L-proline, 7 mL of 1 N NaOH, 25 µg/mL nalidixic acid, 10 µg/mL benomyl, 15 g of agar, and 1 L of double-distilled water. Plates were incubated for 5–7 days at 28 °C. For the strain, 3 mL seed cultures in 14 mL dual-position cap tubes were inoculated with a loopful of vegetative cells from R2YE plates and incubated for 5 days at 28 °C, 200 rpm; 3 mL seed cultures were then inoculated into 100 mL seed cultures in 250 mL baffled flasks and incubated for 7 days at 28 °C, 200 rpm; 50 mL of seed cultures were inoculated into 1 L of fermentation media in 2.8 L baffled Fernbach flasks and grown for 7 days at at 28 °C, 200 rpm.

AT day 7 of the fermentation, 25 g of Amberlite XAD16 resin contained within a polypropylene mesh bag was added to each fermentation culture and agitated overnight at at 28 °C, 200 rpm. On day 8, all resin bags were removed and thoroughly washed with deionized water to remove any water-soluble media components and residual cell mass adsorbed on the resin bags. Each washed resin bag was extracted with 250 mL of MeOH and 250 mL of ethyl acetate (EtOAc). The combined organic fractions were dried under vacuum and redissolved in a minimal amount of MeOH. The solutions were then centrifuged, and the supernatants were loaded onto C18 resin and dried under vacuum prior to Biotage C18 fractionation.

Purification of 2-aminobenzamide-actiphenol. Preparative RF-HPLC fractionation was performed using the same Shimadzu LC-20AP system equipped with a reverse-phase Phenomenex Kinetex[®] 5 mm C18 100 Å (250 × 21.2 mm) column. The materials of strain *Streptomyces actiphen* (brought up in methanol at approximately ~100 to 200 mg/mL) were eluted with a flow rate of 20 mL/min and a linear gradient starting with a 2 min isocratic wash step using 10% acetonitrile/H₂O (with 0.01% TFA), then a 30 min linear gradient step from 10% acetonitrile/H₂O (with 0.01% TFA) to 100% acetonitrile/H₂O (with 0.01% TFA), and then a 5 min wash with 100% acetonitrile/H₂O (with 0.01% TFA) followed by an 8 min equilibration with 10% acetonitrile/ H₂O (with 0.01% TFA). Less than 0.05 mg of the compound is isolated per liter of broth.

Preparative HPLC (Prep HPLC) HF16–18 was fractionated using 40 minutes of an isocratic method with 33% acetonitrile solvent B in H₂O (both solvents A/B with 0.01% TFA) elution. The following compounds were purified using the same semi-preparative system with a reverse-phase Phenomenex Luna[®] 5 mm Phenylhexyl 100 Å (250 × 10 mm) column. 2-Aminobenzamide-actiphenol (approximately 0.05 mg/L) was purified from HF16–18_F24–28 using 30 minutes of an isocratic 35% acetonitrile/H₂O (with 0.01% TFA) elution.

Compound Entries. 2-Aminobenzamide-actiphenol: yellow solid; yellow solid; UV (MeOH) λ_{max} (log ε) 218 (8.9), 267 (6.1), 355 (3.35) nm; ¹H and ¹³C NMR data (Supplementary Table 3); HRESIMS ion peak of C₂₂H₂₂N₃O₅ [M-H]⁻ (*m/z*: found 408.1455, calcd 408.1559).

Cell-Based Bioactivity Analysis. HCT15 (CCI-225) and SW-48 (CCL-231) cells were purchased from ATCC. All cell lines were Mycoplasma free and independently authenticated by short tandem repeat profiling, performed by ATCC. Cells were grown and cultured according to ATCC recommendations. HCT15 cells were cultured in RPMI1640 (30-2001) supplemented with 10% FBS (30-2020). SW48 cells were cultured in Leibovitz's L-15 medium (30-2008) containing 10% FBS. SW-48 cells were grown and treated in an incubator set for atmospheric conditions (no supplemental CO₂ addition). For cell-based assays, cells were expanded and frozen into single-use aliquots. For each assay, cells were thawed at °C

for 1 minute and then immediately re-suspended in 10ml of complete growth medium. Cells were then spun down at 300Xg for 5 minutes and then resuspended in cell-specific growth medium and plated at 2,500 cells per well into Greiner 781080 white cell culture 384-well plates with a total volume per well at 40µl. Natural product extracts or fractions were dissolved in DMSO at 15mg/ml and delivered into the assay plates using Echo 655 acoustic liquid handler instrumentation (Beckman Coulter). Extract and fraction testing concentrations were at 0.25%. For primary screening assays, extract testing was performed n=1 at 0.25% final extract testing concentration (where the original fraction is defined at 100%). Validation assay and fraction studies were performed in triplicate at similar testing concentrations. Negative controls (medium only plus matching 0.25% DMSO) were included in columns 1 and 2. The positive control for these studies was a 10 µM treatment with staurosporine in columns 23 and 24 of each assay plate. Samples were interrogated in wells A03 to P22. The high-throughput data software Mscreen was utilized for primary hit, validation selection, and analysis of concentration-response curve results [<https://doi.org/10.1177/1087057112450186>]. Following compound addition, cells were cultured for 48 hours at either 5% CO₂ at 37°C for HCT15 cells or atmospheric air at 37°C for SW-48 cells. Cell viability was measured using a CellTiter-Glo luminescent kit (catalog no. G7571) from Promega as directed using a PHERAstar instrument from BMG Labtech.

Below sentence is added to the discussion section to conclude this finding:

Actiphenol, a prominent member of the glutarimide-containing polyketide family, has long been recognized for its role as a eukaryotic translation inhibitor. Seq2PKS unveiled a new variant of actiphenol, named 2-aminobenzamide-actiphenol from *Streptomyces actiphen*. This molecule diverges notably from classical actiphenol structures by lacking a methyl group at C-4 and introducing both a new methylene group and an aminobenzamide moiety. Although initial tests revealed promising cytotoxic effects in crude extracts against colon cancer cell lines, the isolated variant did not exhibit substantial activity in further assays. Further studies are needed to explore the bioactive properties of this new compound.

C 3.6 Given the availability of existing structure prediction programs, it is not entirely clear what unique advantages the new pipeline offers.

R3.6. Seq2PKS improves on current existing programs PRISM and antiSMASH in a couple of aspects. In the case of PRISM, we only have access to overall accuracy, but for other methods, we have highlighted step-by-step accuracy benchmarks. Overall, Seq2PKS achieves 94.4% accuracy, which is comparable to antiSMASH in AT-domain specificity prediction. However, Seq2PKS improves prediction accuracy for AT-domain specificity by using an extra-tree-based model optimized to achieve high accuracy for test data points that are far from training data. To measure distance between test data points and training data points, we used Hamming distance on the 24 amino acid signatures [[https://doi.org/10.1016/S0022-2836\(03\)00232-8](https://doi.org/10.1016/S0022-2836(03)00232-8)]. For testing samples that are at least six Hamming distances away from any training data points, Seq2PKS achieve accuracy of 51%, compared to 38% for antiSMASH (Figure 2(a)). During the assembly line order prediction process, Seq2PKS employs a logistic regression algorithm, taking the docking domain sequences as input to predict the optimal assembly pathway for candidate core structures. This method achieves 92% accuracy for non-co-linear polyketide BGCs, while DDAP and antiSMASH achieve 55% and 74% accuracy, respectively. Seq2PKS also improves on existing methods by applying more comprehensive post-assembly modifications. Overall, Seq2PKS predicts 8 out of 80 polyketides in the benchmark dataset from the PRISM paper correctly (16 if counting polyketides outside PRISM dataset), while PRISM and antiSMASH do not predict any of the polyketides correctly. The highest Tanimoto similarity that PRISM and antiSMASH achieve are 0.65 and 0.30.

Figure 2(a): The accuracy of different classifiers for predicting the substrate specificity of AT-domains. The notation B_k+ represents the bin containing data points at least k Hamming distances away from any training data points. While most methods achieve similar accuracy, the extra tree classifier generalizes better, i.e., it achieves higher accuracy for testing samples that are dissimilar to the training samples.

We added the following sentence to the discussion:

Compared to other methods, in AT-domain specificity prediction, Seq2PKS achieves over 94% accuracy, comparable to the accuracy for antiSMASH. However, Seq2PKS improves prediction accuracy for AT-domain specificity by using an extra-tree-based model optimized to achieve high accuracy for test data points that are far from training data. For testing samples that are at least six Hamming distances away from any training data points, Seq2PKS achieves an accuracy of 51%, compared to 38% for antiSMASH. During the assembly line order prediction process, Seq2PKS employs a logistic regression algorithm, taking the docking domain sequences as input to predict the optimal assembly pathway for candidate core structures. This method achieves 92% accuracy for non-co-linear polyketide BGCs, while DDAP and antiSMASH achieve 55% and 74% accuracy, respectively. Furthermore, Seq2PKS uses a subgraph isomorphism algorithm to consider a more comprehensive set of post-assembly modifications. Overall, Seq2PKS correctly predicts eight out of 80 polyketides in the benchmark dataset (16 if counting polyketides without PRISM result), while PRISM and antiSMASH do not predict any of the polyketides correctly. The highest Tanimoto similarity that PRISM and antiSMASH achieve are 0.65 and 0.30.

C 3.7 Furthermore, it is crucial to consider the substantial number of potential structures that can be

generated from a single structural template and assess the potential for false-positive identifications in spectral data.

R3.7. We have added the below analysis to the method section to assess the false-discovery rate in the spectral data.

Estimating False Discovery Rate. The false discovery rate (FDR) is estimated with a target-decoy approach. We select nystatin as the testing molecule. The target database is generated by applying all the detected modifications to the correct core structures. Decoys are generated via a fixed number of edge-switching steps on target database molecules. The edge-switching operation selects two edges in a graph and randomly swaps their endpoints. To operate on molecules, we model a molecule as a connected multigraph and enforce that there is only a single connected component after an edge-switching operation. We generated decoys for the target database by applying 25 edge switching steps on each compound and filtering out generated decoys that appear in the target database by InChIKey. The p-value for all the matching in the decoy and target database is shown in Supplementary Figures 22 and 23. Given a p-value threshold, let N_{decoy} and N_{target} denote the number of peptide-spectrum matches in the decoy database and the target database, respectively. The FDR can be estimated as follows:

$$FDR = N_{decoy} / N_{target}$$

Supplementary Figure 22: Number of polyketide-spectrum matches/peptides identified by Seq2PKS at different p-value thresholds for the target and decoy databases for Nystatin. The X-axis represents different p-value thresholds, and the Y-axis represents the number of identified polyketide-spectrum matches or peptides.

Supplementary Figure 23: False discovery rate of Seq2PKS at different p-value thresholds for the target and decoy databases for Nystatin. X-axis represents different p-value thresholds, and Y-axis represents the corresponding false discovery rate.

C 3.8 Unfortunately, an attempt to test the server, as recommended on the GitHub page, was hindered by its unavailability.

R3.8. We apologize about this. The server is now accessible at <https://run.npanalysis.org/>. Guidelines for using the server are available at <https://github.com/mohimanilab/Seq2PKS> and shown below. The sample data for testing the server is attached and provided in the GitHub page.

The Seq2PKS pipeline requires paired genomic and mass spectrometry data. After getting to the landing page, proceed to the Dashboard.

NP Analysis

Dashboard

Account

In-Silico Tools for Natural Product Analysis

HypoRiPPAtlas	VInSMoC	NPDdiscover	Seq2PKS
---------------	---------	-------------	---------

First, the user will need to create the user account.

Once the user logs in, they will be directed to the Dashboard page, where they should upload genome data and paired mass spectrometry data. Currently, we do not support Seq2PKS runs without paired mass spectrometry data, due to the high costs associated with large data streaming. To upload data, click on the Data tab, click the plus icon, select the type of data, and upload the file. The contents of the data are processed for use in downstream analysis, and the raw files are not used after this processing step.

ID	Name	Created
----	------	---------

Dashboard

Projects **Data** Tasks

Account

ID	Name	Kind	Created	Status
----	------	------	---------	--------

Dashboard

Projects **Data** Tasks

Account

Upload New Data

Upload Data

Data Kind

Mass Spectrum

Upload Your Data

Select Files

chalomycin_specific.mzXML

You may upload MGF, mzXML, or mzML files

Submit

After uploading the data, it will appear under the Data tab.

ID	Name	Kind	Created	Status
90239696-8fd8-4658-a4a4-09a1823a4c4d	chalconmycin_specific.mzXML	Mass Spectrum	Mar 10, 2024 @ 20:06:30	Success
0403075c-cb53-4106-965b-c51523d93677	chalconmycin.fasta	Genome	Mar 10, 2024 @ 20:06:30	Success

To start a new PKS workflow, the user can click on the dashboard tab. To create a new project, click the plus icon.

ID	Name	Created
----	------	---------

Next, to use the uploaded data in a project, the user should click on the project they created and link the data to the project.

NP Analysis

Dashboard

Account

Share

Delete

ID: 47160ee8-db8c-45eb-90a7-5e499ebf9b5b

Created: Mar 10, 2024 @ 19:55:38

Tasks

ID	Name	Method	Created	Completed	Status

Data

ID	Name	Kind	Created	Status
622e1903-7248-458e-8da4-0258ff9b40cd	NPAtlas	Chemical Database	Mar 10, 2024 @ 19:55:46	Success

Upon linking the data, it will appear in the Data section under the project. Now the user can submit a new Seq2PKS task using their data

NP Analysis

Dashboard

Account

ID: 47160ee8-db8c-45eb-90a7-5e499ebf9b5b

Created: Mar 10, 2024 @ 19:55:38

Tasks

ID	Name	Method	Created	Completed	Status

Data

ID	Name	Kind	Created	Status
90239696-8fd8-4658-a4a4-09a1823a4c4d	chalconycin_specific.mzXML	Mass Spectrum	Mar 10, 2024 @ 19:57:08	Success
0403075c-cb53-4106-965b-c51523d93677	chalconycin.fasta	Genome	Mar 10, 2024 @ 19:57:08	Success
622e1903-7248-458e-8da4-0258ff9b40cd	NPAtlas	Chemical Database	Mar 10, 2024 @ 19:57:08	Success

Here the user can input several different parameters, which are described below:

Number of Filtered Cores	A hard maximum on the number of PK core structures to consider in downstream analysis.
Ion Mode	Possible ion modes for the input spectra, either positive or negative.
Precursor Ion Tolerance	Absolute error tolerance when matching precursor ions to molecules.
Product Ion Tolerance	Absolute error tolerance when matching peaks to theoretical fragments.
Max Absolute Charge	The maximum absolute charge, used to pick adducts.
Minimum Score	The minimum score for a molecule-spectrum match to be considered. Hits with scores lower than this will be silently discarded.

To submit the task, scroll down and click the Submit button. After the task is completed, the user can click on the task ID in the Tasks section of the project page to view the results of the task.

✕ NP Analysis
👤

Dashboard
ID 13fb3a02-22e4-4b73-966e-f268cd24e99a

Account
Method Seq2PKS

Processing Status
Finished

Created
Mar 10, 2024 @ 19:58:00

Completed
Mar 10, 2024 @ 19:58:15

Results

Items per page: 10 ▾
1-1 of 1 item
1 ▾ of 1 page
◀ ▶

Molecule	Scan Number	Precursor m/z	Retention Time	Molecule Exact Mass	Mass Error	Adduct	Score
<chem>CC1C(OC4C(C(OC)C(O)C(C)O4)OC)C=CC3C(O3)C(=O)C(CC(C(OC2OC(C)CC(C2O)OC)C(C(=O)CC(O1)=O)C)C)C</chem>	5485	701.370	3194.650	700.367	-0.004	[M+H] ⁺	25

The results presented above were generated using the sample Chalcomyacin BGC sequence and the corresponding mass spectrometry data as inputs. Given that this was conducted for testing purposes, the mass spectrum included only one scan that matched to Chalcomyacin.

REVIEWERS' COMMENTS

Reviewer #1 (Remarks to the Author):

The authors have considerably revised their manuscript and addressed the issues identified in the initial version. They now have included references and discussions to previous tools using similar approaches where appropriate. A webserver providing the tool and an user guide now is now available. A docker container furthermore should ease the installation of the complex pipeline on own computing infrastructure.

Reviewer #2 (Remarks to the Author):

My concerns had been fully addressed.

Reviewer #3 (Remarks to the Author):

All my comments and concerns have been adequately addressed. I believe that the manuscript has been much approved and it is now clear what advantages the new program provides. The figures and method descriptions also have been improved.